# GROKKING AS THE TRANSITION FROM LAZY TO RICH TRAINING DYNAMICS

**Tanishq Kumar**  **Blake Bordelon**  **Samuel J. Gershman**[*]  **Cengiz Pehlevan**[*]
Harvard University

## ABSTRACT

We propose that the grokking phenomenon, where the train loss of a neural network decreases much earlier than its test loss, can arise due to a neural network transitioning from lazy training dynamics to a rich, feature learning regime. To illustrate this mechanism, we study the simple setting of vanilla gradient descent on a polynomial regression problem with a two layer neural network which exhibits grokking without regularization in a way that cannot be explained by existing theories. We identify sufficient statistics for the test loss of such a network, and tracking these over training reveals that grokking arises in this setting when the network first attempts to fit a kernel regression solution with its initial features, followed by late-time feature learning where a generalizing solution is identified after train loss is already low. We find that the key determinants of grokking are the rate of feature learning—which can be controlled precisely by parameters that scale the network output—and the alignment of the initial features with the target function $y(x)$. We argue this delayed generalization arises when (1) the top eigenvectors of the initial neural tangent kernel and the task labels $y(x)$ are misaligned, but (2) the dataset size is large enough so that it is possible for the network to generalize eventually, but not so large that train loss perfectly tracks test loss at all epochs, and (3) the network begins training in the lazy regime so does not learn features immediately. We conclude with evidence that this transition from lazy (linear model) to rich training (feature learning) can control grokking in more general settings, like on MNIST, one-layer Transformers, and student-teacher networks.

## 1  INTRODUCTION

The goal of a machine learning system is to learn models that generalize beyond their training set. Typically, a practitioner hopes that a model's performance on its training set will be an indicator of its generalization capabilities on unseen data. However, this may not happen in practice. Grokking, discovered by Power et al. (2022), is a phenomenon where the train loss of a network falls initially with no corresponding decrease in test loss, then the network generalizes later during training. There has been much recent work studying this phenomenon not only because it challenges the widely-used approaches of early stopping or train loss termination criteria, but also because some Nanda et al. (2023) see grokking as an example of sudden, emergent behavior in a neural network, and a principle aim of the field of ML interpretability is to understand model internals well enough that they do not exhibit such unexpected and unpredictable changes in behavior.

For these reasons, the phenomenology of grokking has been under close empirical study since its discovery (Nanda et al., 2023; Davies et al., 2023; Thilak et al., 2022; Varma et al., 2023; Liu et al., 2022a;b; Gromov, 2023). A limited number of theories have been proposed to explain the causes of grokking, most notably those that attribute grokking to weight decay and weight norm decrease (Liu et al., 2022a; Varma et al., 2023). However, we give counterexamples to such theories, implying a need to explain grokking in a way that is consistent with empirical observations in current and past work.

In this paper, we propose the idea that grokking is caused by delayed feature learning during training. Our theoretical framework relies on the idea of lazy training where a network can fit its training

---

[*]Equal Senior Authors. Correspondence to cpehlevan@seas.harvard.edu.

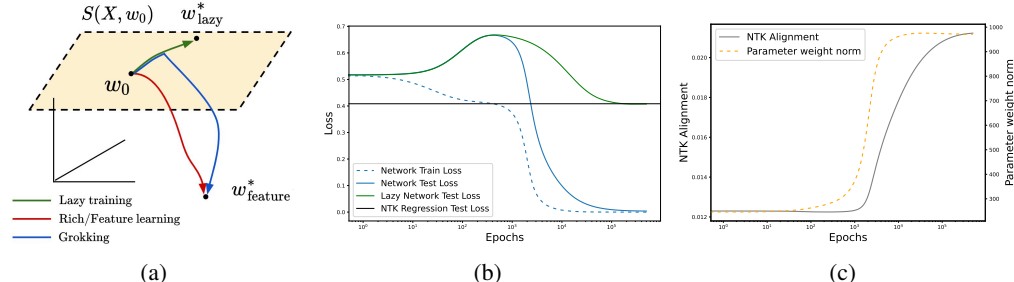

(a)  (b)  (c)

Figure 1: (a) Claimed parameter dynamics during grokking in a parameter space of $\mathbb{R}^3$ for illustrative purposes. $S(X, w_0)$ is an affine subspace of parameter space reachable by models linearized around $w_0$. (b) Grokking on a polynomial regression task introduced in Section 5 with an MLP, vanilla GD, and zero regularization. Green and blue loss curves in (b) correspond to sketched green and blue parameter dynamics in (a). Horizontal line (black) in (b) is the mean-squared error of best kernel regression estimate with the NTK at initialization. (c) The fact that parameter weight norm increases (dashed orange) cannot be explained by any existing theories of grokking. Features get aligned (grey) in a way we make precise in Section 5.

data without adapting its internal representations (Chizat et al., 2019). In this regime, the network behaves as a linearized model with features evaluated at initialization (Jacot et al., 2018; Chizat et al., 2019; Lee et al., 2019)[1]. We show that grokking can arise when a network begins fitting and "memorizing" its training set (albeit imperfectly) in the linearized regime, then later adapts its features to the data, leading to improved generalization at late time.

We illustrate this idea schematically in Figure 1(a). Suppose we randomly initialize a network with the parameters $w_0$. When linearized around $w_0$, under gradient descent, a neural network's weights are restricted to an affine subspace spanned by initial gradients $w_0 + \text{span}\{\nabla_w f(w, x_\mu)|_{w_0}\}_{\mu=1}^P$, and gradient descent finds a training minimizer on this subspace, $w_{\text{lazy}}$. For MSE loss this minimizer is the kernel regression solution with the network's initial Neural Tangent Kernel (NTK) (Jacot et al., 2018; Lee et al., 2019). Alternatively, if the network leaves the linearized regime, the weights can leave the subspace of the linear model and converge to another solution $w_{\text{feature}}$. We hypothesize that grokking can occur when a network transitions from initially tracking the linearized network dynamics in the subspace to more complicated dynamics that approach $w_{\text{feature}}$ off of the subspace which achieves better generalization because it learns features.

Given a fixed dataset where grokking is possible, our proposal naturally suggests two key factors that control grokking: (1) how much feature learning is necessary for good performance, and (2) how fast these features are learned. We introduce two parameters corresponding to these notions and demonstrate they can control grokking.

The key contributions in this paper are the following:

- We give simple examples of a two-layer MLP without any weight decay that exhibits grokking in ways inconsistent with existing theories of grokking.

- We demonstrate with both theory and experiments that grokking in this setting is associated with the transition from lazy training to a rich, feature learning regime, and can be tuned by parameters that control this transition. These parameters are:

  – A scale parameter, $\alpha$, that controls the magnitude of the output of the network, and thus the rate of feature learning throughout training (ie. laziness).
  – The task-model kernel alignment between the neural tangent kernel (NTK) of the network at initialization and the test labels, a measure of how much feature learning is necessary for good performance on the task (Cortes et al., 2012).

---

[1]We use the terms "lazy," "kernel regime" and "linearized model" interchangeably to refer to the same phenomenon where the network learns without changing its internal features. Concretely, this is a regime where the network output $f$ can move a large amount while $\nabla_w f$ stays approximately constant.

- A "goldilocks" train set size so that generalization is possible but not immediate (for instance, grokking is impossible in the infinite data limit where train loss perfectly tracks test loss).

- We provide multiple lines of evidence that the feature learning dynamics that give grokking in this simple setup are also those at play during grokking in more complicated settings studied in past work: on MNIST, with one-layer transformers, and with student-teacher networks. Given a set of training inputs, we give a method to construct labels that can generate grokking.

## 2 RELATED WORK

**Grokking.** Nanda et al. (2023) mechanistically interpret the algorithm learned during grokking for modular addition tasks. We show in Section 6 that merely tuning the network output scaling/laziness parameter $\alpha$ is alone sufficient to make grokking continuously vanish on the one-layer Transformer used in that work. Thilak et al. (2022) argue that grokking is caused by an optimization anomaly of adaptive optimizers. We find grokking with vanilla gradient descent, so this cannot be the cause of grokking in the general case. While we show in experiments in 6 that parameters from our theory control grokking with adaptive optimizers, we note in Appendix Section 14 that learning dynamics in such settings are less well understood. Davies et al. (2023) point out that grokking and double descent share similarities in dynamics, and are fundamentally about the speed at which patterns of varying complexity are learned. This agrees with our argument that the network first learns patterns in the data that a linearized model can do well on, then changes its features to learn the patterns that generalize. Liu et al. (2022a) empirically find that parameter weight norm at initialization can control grokking, while Varma et al. (2023) propose a heuristic for cross-entropy training with weight decay, suggesting that grokking arises from a "generalizing circuit" being favored over a "memorizing circuit." Both papers rely on weight decay and weight norm decrease to explain grokking. However, we show examples of grokking with zero weight decay and an *increase* in parameter weight norm during training. We comment more on relations to existing theories in Appendix Section 9.

**Kernel dynamics and feature learning in a polynomial setting.** The limiting behavior of neural networks in the large width limit (under commonly used parameterizations) is a linear model with the NTK as the kernel (Jacot et al., 2018). This has motivated investigations into the inductive biases of kernel methods (Bordelon et al., 2020; Canatar et al., 2021), which have revealed that kernels are strongly biased to explain training data using the top components of their eigenspectrum. While this spectral bias can enable good generalization on some learning problems, target functions that are not in the top eigenspaces of the kernel require a potentially much larger amount of data. Many papers have proposed examples where neural networks can outperform kernels statistically by amplifying task-relevant dimensions compared to task-irrelevant dimensions in the input space (Mei et al., 2018; Paccolat et al., 2021; Refinetti et al., 2021). One example of such a problem that has attracted recent attention is single or multi-index models where the target function is a polynomial that depends on a small subset of the high dimensional inputs (Ba et al., 2022; Arnaboldi et al., 2023; Nichani et al., 2022; Atanasov et al., 2022; Berthier et al., 2023; Bietti et al., 2022). The fact that feature-learning neural networks can outperform kernels on these tasks suggests that they could be useful toy models of grokking, especially if early training is close to linearized dynamics.

## 3 WEIGHT NORM AND WEIGHT DECAY CANNOT EXPLAIN GROKKING

Previous explanations of grokking rely on weight norm decreases at late time (Liu et al., 2022a; Varma et al., 2023). In Figure 2, we show a simple example of a modular arithmetic task, which is a common task under study in grokking (Power et al., 2022; Nanda et al., 2023; Gromov, 2023), with a two-layer MLP trained without any weight decay (details in Appendix Section 8.3). Weight norm of parameters *increases* during training (Figure 2(a)), so grokking cannot in general be explained by theories of weight decay.

We will focus on the dynamics of grokking in the *loss* (Figure 2(b)), because loss, not accuracy, guides training dynamics. Further, we show in Section 19 of the Appendix that accuracy curves for *regression* tasks are misleading because they can be gamed by choice of metric. We use mean-squared error (MSE) loss on this classification task to adhere to the convention for how grokking was seen in modular arithmetic in Liu et al. (2022a); Gromov (2023). We show in Section 6 that our results remain consistent across architectures, optimizers, and datasets out of the box.

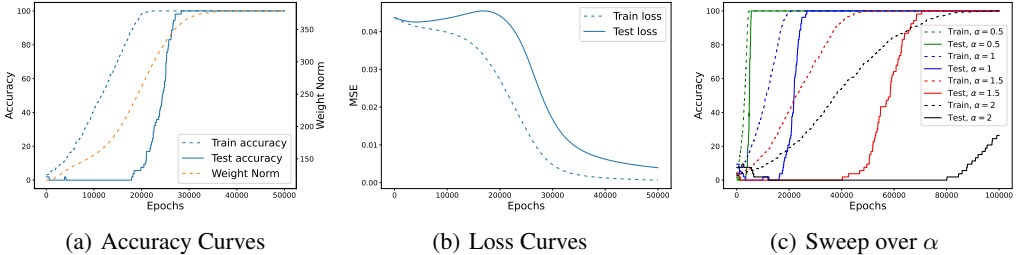

(a) Accuracy Curves        (b) Loss Curves        (c) Sweep over $\alpha$

Figure 2: (a) and (b) demonstrate accuracy and loss curves for grokking on a modular arithmetic task that shows an *increase* in parameter weight norm during training. (c) is a sweep over the laziness parameter $\alpha$ that we will introduce and provide a theory for, showcasing how it can continuously control grokking.

While we gave the modular arithmetic task as a counterexample first because it is a common task under study in grokking, we will focus on a polynomial regression task in the rest of the paper, which we can analyze the dynamics of mathematically. We introduce that task in Section 5. Parameter weight norm in that example *also increases* as it groks, as visible in Figure 1 (c). These inconsistencies motivate our study which results in two key parameters – network laziness and NTK alignment – that we introduce. The effects of varying the first, $\alpha$, are shown in Figure 2(c) and we see it can make grokking more dramatic, or vanish entirely.

## 4 GROKKING AS TRANSITION FROM LAZY TO RICH LEARNING

Grokking is often conceptualized as a network that first finds a memorizing solution which fits the training set but not the test set, later converging to a generalizing solution. We hypothesize that the "memorizing" solutions that have been documented in prior works on grokking are equivalent to *early-time lazy training dynamics*. At late time, the linearized approximation breaks down as the network extracts useful features and begins to "grok."

We next briefly review and define the idea of lazy training regime of neural networks, and introduce some key theoretical concepts that will play an important role in our discussion.

### 4.1 PRIMER ON LAZY LEARNING: LINEAR VS NONLINEAR TRAINING REGIMES

In the lazy training regime, a network $f(\boldsymbol{w}, \boldsymbol{x})$ of inputs $\boldsymbol{x}$ and parameters $\boldsymbol{w}$ obey the approximation

$$f(\boldsymbol{w}, \boldsymbol{x}) \approx f(\boldsymbol{w}_0, \boldsymbol{x}) + \nabla_{\boldsymbol{w}} f(\boldsymbol{w}, \boldsymbol{x})|_{\boldsymbol{w}_0} \cdot (\boldsymbol{w} - \boldsymbol{w}_0). \tag{1}$$

While this approximation is still capable of learning nonlinear functions of the input $\boldsymbol{x}$, it is a *linear model* in the trainable parameters $\boldsymbol{w}$. It can thus be recast as a kernel method with the neural tangent kernel $K$

$$K(\boldsymbol{x}, \boldsymbol{x}') = \nabla_{\boldsymbol{w}} f(\boldsymbol{w}, \boldsymbol{x})|_{\boldsymbol{w}_0} \cdot \nabla_{\boldsymbol{w}} f(\boldsymbol{w}, \boldsymbol{x}')|_{\boldsymbol{w}_0} \tag{2}$$

evaluated at initialization. This holds for any linearized model trained with GD on any loss, though some loss functions (e.g. cross-entropy) cause non-linear models to eventually deviate from their linearization (Lee et al., 2019). Deep neural networks can approach this linearization in many ways:

- Large width: in commonly used parameterization and initialization schemes of neural networks, increasing the network width improves the quality of this approximation Jacot et al. (2018); Lee et al. (2019).

- Large initial weights: increasing the size of the initial weights also causes the network to behave closer to linearized dynamics (Chizat et al., 2019). *We argue that this explains why Liu et al. (2022a) and Varma et al. (2023) observed an association between grokking and large initial weight norm.*

- Label rescaling: scaling $y$ by factor $\alpha^{-1}$ can induce lazy training (Geiger et al., 2020).
- Output rescaling: multiplying the network output logits by a large scale factor $\alpha$ can also induce lazy training (Chizat et al., 2019).

In this work, we primarily use the last option, where the $\alpha \to \infty$ limit is one where equation 1 becomes exact and the network behaves at inference-time like a linear model (in the case of MSE, this linear model is kernel regression with the initial NTK). In Figure 7, we show that these methods of inducing laziness are equivalent in inducing grokking.

**Inductive bias of kernels and deficiency of kernels on misaligned tasks:** Though linearized models around the initial weights can perform well on some learning tasks, they do not generalize well when trained on target functions $y(x)$ which are *misaligned* to the NTK, in a sense we make precise when we define centered-kernel alignment (CKA) in the next section. By contrast, neural networks in the feature learning regime can adapt their internal representations to improve the sample complexity of learning for functions that would be difficult for the NTK (Ghorbani et al., 2019; Arnaboldi et al., 2023; Ba et al., 2022; Damian et al., 2023).

## 5 POLYNOMIAL REGRESSION IN A TWO LAYER PERCEPTRON

To illustrate our main ideas, we study the simplest possible toy example of grokking: high dimensional polynomial regression. This choice is motivated by known separation results between neural networks and kernel methods. First, theoretical studies of kernel regression have demonstrated that learning degree $k$ polynomials in dimension $D$ with the NTK requires $P \sim D^k$ samples (Bordelon et al., 2020; Canatar et al., 2021; Xiao et al., 2022). On the other hand, in the feature learning regime, two-layer networks trained in the online setting can learn high-degree polynomials with $P \sim D$ samples (Saad and Solla, 1995; Engel, 2001; Goldt et al., 2019; Arnaboldi et al., 2023; Berthier et al., 2023; Sarao Mannelli et al., 2020). This indicates that neural networks can potentially generalize for $D \ll P \ll D^k$, but only if they learn features (Atanasov et al., 2022). We therefore predict the possibility of grokking in this intermediate range of sample sizes, especially if feature learning occurs late in training, as in a lazy network. We indeed find we can induce grokking this way on a two-layer network. In fact, grokking persists even with readout weights of the 2 layer MLP fixed to 1, so in seeking a minimal setting reproducing grokking, this is what we study[2]. This type of two-layer NN is referred to as a committee machine (Saad and Solla, 1995).

The model $f(\boldsymbol{w}, \boldsymbol{x})$ and target function $y(\boldsymbol{x})$ are defined in terms of input $\boldsymbol{x} \in \mathbb{R}^D$ as

$$f(\boldsymbol{w}, \boldsymbol{x}) = \frac{\alpha}{N} \sum_{i=1}^{N} \phi(\boldsymbol{w}_i \cdot \boldsymbol{x}), \quad \phi(h) = h + \frac{\epsilon}{2} h^2, \quad y(\boldsymbol{x}) = \frac{1}{2} (\boldsymbol{\beta}_\star \cdot \boldsymbol{x})^2 \tag{3}$$

The value of $\alpha$ controls the scale of the output, and consequently the speed of feature learning. The value of $\epsilon$ alters how difficult the task is for the initial NTK. We consider training on a fixed dataset $\{(\boldsymbol{x}_\mu, y_\mu)\}_{\mu=1}^{P}$ of $P$ samples. The inputs $\boldsymbol{x}$ are drawn from an isotropic Gaussian distribution $\boldsymbol{x} \sim \mathcal{N}(0, \frac{1}{D}\boldsymbol{I})$. It will be convenient to introduce the following two summary statistics

$$\bar{\boldsymbol{w}} = \frac{1}{N} \sum_{i=1}^{N} \boldsymbol{w}_i \in \mathbb{R}^D, \quad \boldsymbol{M} = \frac{1}{N} \sum_{i=1}^{N} \boldsymbol{w}_i \boldsymbol{w}_i^\top \in \mathbb{R}^{D \times D} \tag{4}$$

Using these two moments of the weights, the neural network function can be written as $f(\boldsymbol{x}) = \alpha \bar{\boldsymbol{w}} \boldsymbol{x} + \frac{\alpha \epsilon}{2} \boldsymbol{x}^\top \boldsymbol{M} \boldsymbol{x}$. Further, the NTK can also be expressed as

$$K(\boldsymbol{x}, \boldsymbol{x}') = \boldsymbol{x} \cdot \boldsymbol{x}' + \epsilon(\boldsymbol{x} \cdot \boldsymbol{x}') \, \bar{\boldsymbol{w}} \cdot (\boldsymbol{x} + \boldsymbol{x}') + \epsilon^2 (\boldsymbol{x} \cdot \boldsymbol{x}') \, \boldsymbol{x}^\top \boldsymbol{M} \boldsymbol{x}' \tag{5}$$

At initialization in a wide network, $\bar{\boldsymbol{w}} = 0$ and $\boldsymbol{M} = \boldsymbol{I}$. Diagonalizing this initial NTK with respect to the Gaussian data distribution reveals that linear functions all have eigenvalue $\lambda_{\text{lin}} = D^{-1}$ while

---

[2]The mechanism behind grokking in this setting is the same for training a full 2 layer MLP with readouts, but the kernel is not only a function of $\bar{\boldsymbol{w}}, \boldsymbol{M}$. We discuss this in Appendix Section 11, but this is a technicality: the same arguments and plots hold in ordinary two-layer MLPs in the way one would expect.

quadratic functions have eigenvalue $\lambda_{\text{quadr}} = 2\epsilon D^{-2}$ (Appendix Section 11). We see that $\epsilon$ controls the power the kernel places in quadratic functions (including the target $y(x)$). This is what we mean by saying $\epsilon$ controls initial alignment in this example. Further, based on prior work on kernel regression, generalization with the initial kernel (or completely lazy neural network) will require $P \sim \lambda_{\text{quadr}}^{-1} = \frac{1}{2\epsilon} D^2$ samples (Bordelon et al., 2020; Canatar et al., 2021; Simon et al., 2023; Xiao et al., 2022). However, neural networks outside of the kernel regime can generalize from far fewer samples since they can adapt their features so that $M$ aligns to $\beta_\star \beta_\star^\top$ and low test error is achieved, which does not require as many samples (Ghorbani et al., 2019; Atanasov et al., 2022). [3]. If this feature learning is delayed, the neural network dynamics at early time would match a kernel method, but at late time one would see generalization as weights begin to align to $\beta_\star$, giving a learning curve that groks. This is the task that generates the learning curves in Figures 1, 3.

While $\epsilon$ is a natural measure of alignment in our toy model, we also want a general way to determine how well aligned a network is for a task, $y(X)$. It turns out that centered-kernel alignment (CKA), $\frac{y^T K_0 y}{||K||_F ||y||^2}$, (Kornblith et al., 2019; Cortes et al., 2012; Atanasov et al., 2022; Arora et al., 2019), is a natural generalization of what $\epsilon$ is in our toy model. In the formula, $K_0$ is the gram matrix of the initial NTK evaluated on the test set, and $y$ are the task labels. We derive a relationship between $\epsilon$ and the CKA in Section 11 of the Appendix, finding they are strongly correlated. Since the CKA can be computed for any task, it is a general way to measure feature alignment to task. "NTK alignment" in plots without further definition refers to the CKA.

## 5.1 ALTERING SCALE $\alpha$ CAN INDUCE OR ELIMINATE GROKKING

We next consider the effect of scale $\alpha$ and show that large $\alpha$ increases the timescale separation between the decrease in train loss and that of test loss, while small $\alpha$ can eliminate grokking altogether. Consider Figure 3(a) to see that increasing $\alpha$ on the polynomial regression task continuously induces a longer delay between train loss decrease and test loss decrease. Further, the $\alpha \to \infty$ limit gives poor final performance since it corresponds to regression with the initial (misaligned) NTK.

**Weight norm of parameters is not inherently fundamental.** A key claim of ours is that previous work Liu et al. (2022a); Varma et al. (2023) saw a relationship between parameter weight norm at initialization and grokking because *weight norm at initialization controls rate of feature learning* by moving the network into the lazy training regime. We can test this claim using the fact that parameter weight norm at initialization is not the only change that can cause lazy training: in the same papers (Chizat et al., 2019; Lee et al., 2019) it is also proved that increasing model output scale or label scale has the same effect. Indeed, as we see in Appendix Figure 7, changing these *has the same effect on learning curves as changing weight norm at initialization*.

## 5.2 INITIAL KERNEL-TASK ALIGNMENT $\epsilon$ CAN ALSO ALTER GROKKING DYNAMICS

The initial alignment between the NTK and the target function also controls grokking behavior. Indeed in Figure 3(b), we see that small $\epsilon$ networks have a longer time delay between training loss reduction and test loss reduction and that large $\epsilon$ networks have a nearly immediate reduction in test loss. Interestingly, worse initial alignment caused by smaller $\epsilon$ leads to lower final test loss as it forces the network to learn features (see App. 10). This means that networks with *bad initial features* can do better at end-time than those with *better initial features*, because they are forced to learn good features.

## 5.3 LOSS DECOMPOSITION IN TOY MODEL

To gain insight into the sources of generalization error in our model, we decompose the test risk into three terms that depend only on the summary statistics $(\bar{w}, M)$ introduced earlier. The predictor can be expressed in terms of these objects: $f = \alpha \bar{w} \cdot x + \frac{\alpha\epsilon}{2} x^\top M x$. The test MSE of our model then

---

[3]Prior work suggests learning quadratic polynomials (with no linear component in $\beta_\star$ direction) with SGD require $\sim \mathcal{O}(D \log D)$ steps in SGD with unit batchsize (Arous et al., 2021).

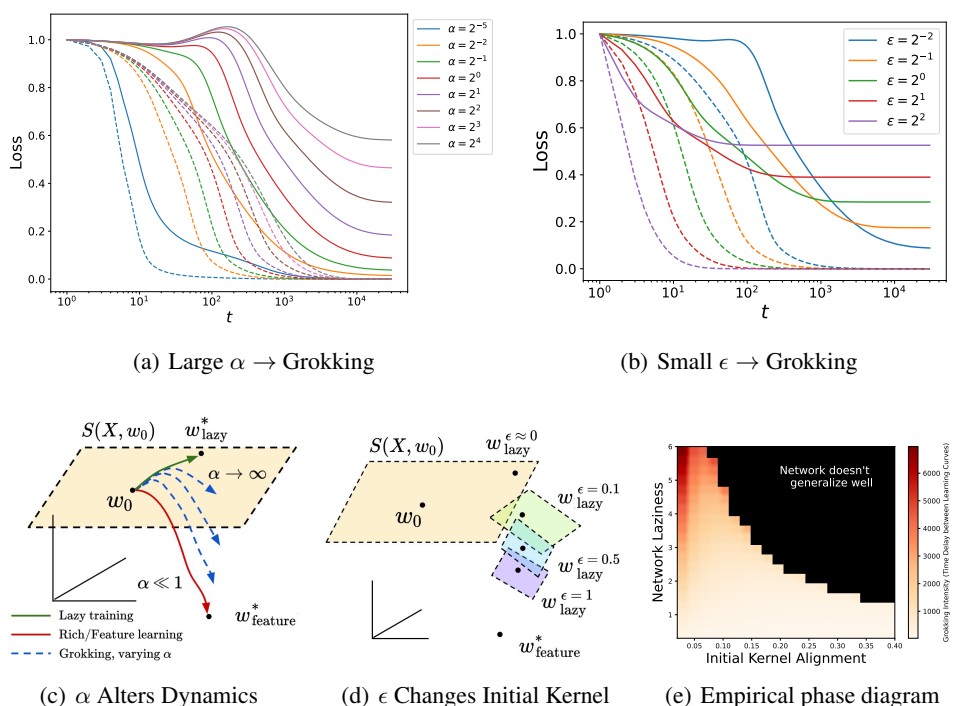

Figure 3: Lazy training ($\alpha$) and kernel-task misalignment ($\epsilon$) alter the grokking learning curves in distinct ways. (Top) Learning curves that show grokking, and (Bottom) corresponding parameter dynamics during learning. (a) At fixed $\epsilon$, the laziness parameter $\alpha$ controls the timescale of the delay in grokking. At small $\alpha$, the grokking effect disappears as the generalizing features are extracted immediately. At large $\alpha$, the model approaches a linearized model. The final test loss decreases with $\alpha$ as we allow the network to learn more features. (b) The task alignment to the initial kernel, measured by $\epsilon$, determines how much the loss falls when the network initially uses its linearized solution. Smaller $\epsilon$ increases the amount of feature learning during training because the initial kernel does worse on the task, so feature learning is necessary. Thus lower alignment can result in better generalization. (c)-(d) Illustrations of the dynamics at varying $\alpha, \epsilon$. In (d), each plane represents a different affine space spanned by the initial gradients, which are a function of $\epsilon$. (e) Time to grok (time delay between train loss fall and test loss fall) as a function of $\alpha, \epsilon$, showing how lazy, misaligned networks grok the most intensely.

takes the form:

$$\mathcal{L} = \left\langle (y-f)^2 \right\rangle = \frac{1}{4} \underbrace{\left( \frac{1}{D} |\boldsymbol{\beta}_\star|^2 - \frac{\alpha\epsilon}{D} \mathrm{Tr} \boldsymbol{M} \right)^2}_{\text{variance error}} + \frac{1}{2D^2} \underbrace{\left| \alpha\epsilon\boldsymbol{M} - \boldsymbol{\beta}_\star\boldsymbol{\beta}_\star^\top \right|_F^2}_{\text{misalignment error}} + \underbrace{\frac{\alpha^2}{D} |\bar{\boldsymbol{w}}|^2}_{\text{power in linear modes}} . \quad (6)$$

The first term measures whether the total variance in the weights $\frac{1}{DN}\sum_i |\boldsymbol{w}_i|^2 = \frac{1}{D}\sum_{j=1}^D M_{jj}$ has the correct scale compared to the targets. The second term measures whether the weights have achieved a solution with *high alignment* to the target function. It is closely related and strongly correlated to its general counterpart, the CKA, $\frac{y^T K_0 y}{||K_0||_F ||y||}$, in a way we explore in Appendix Section 11. The last term penalizes any power our features put on linear components, because our target function is a pure quadratic. Because our network activation function has a linear component, weight vectors must average out during training to learn a solution with zero overall power in linear functions. The test error can be minimized by any neural network satisfying $\boldsymbol{M} = \frac{1}{\alpha\epsilon}\boldsymbol{\beta}_\star\boldsymbol{\beta}_\star^\top$ and $\bar{\boldsymbol{w}} = 0$.

In Figure 4 we illustrate this decomposition of the loss during a grokking learning curve. We find, as expected, that the initial peak in test error is associated with the model attempting to learn a linear function of the input (orange curve) before later improving the alignment (dashed green loss falling at late time) and reducing the scale of the linear component (dashed black).

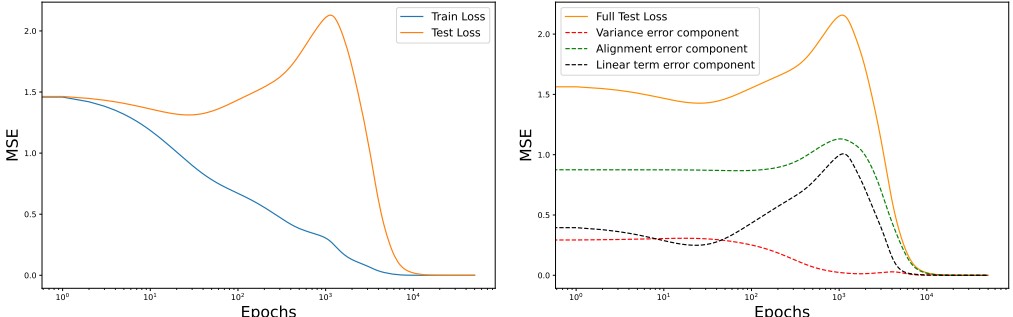

Figure 4: (Left) Learning curves for a 2 layer multi-layer perceptron that groks on our polynomial regression task (no weight decay, vanilla GD). (Right) Theoretical decomposition showing how the initial rise in test loss comes from the network putting power in the linear component from its inital NTK before beginning to align with the task features around 1-2k epochs, resulting in a delayed fall in the test loss. This delay is precisely grokking.

# 6 GROKKING IS NOT RESTRICTED TO MODULAR ARITHMETIC

## 6.1 INDUCING GROKKING ON GAUSSIAN DATA BY LOWERING ALIGNMENT OF LABELS

In this subsection, we induce grokking by manipulating NTK alignment at initialization in a way that can be done on arbitrary datasets $X$, without touching the laziness parameter. Consider the same function as in our polynomial regression setup but replace the task labels with the $j$-th largest eigenvector (by descending eigenvalue, so $j = 1$ is the eigenvector of $K_0(X, X)$ with highest eigenvalue) of the NTK matrix at initialization $K_0(X, X)$, and vary $j$. This is motivated because choosing an eigenvector with a large eigenvalue makes the CKA, $\frac{y^T K_0 y}{||K_0||_F ||y||^2}$, high by construction, and vice versa. Thus alignment at initialization is high for small $j$ (first plot), so the network generalizes immediately because it starts with good features. Conversely, if we begin with large $j$ the value of CKA is low, and the task is hard by construction, so the network will never be able to generalize with this amount of data (with much more data, we would learn features). Intermediate $j$ eigenfunctions are misaligned *enough* to the initial NTK that a linearized model will not perform well, but the network has enough data so that learning features reduces the test loss. We see empirically that this occurs when $j \approx 70$ in (b) below.

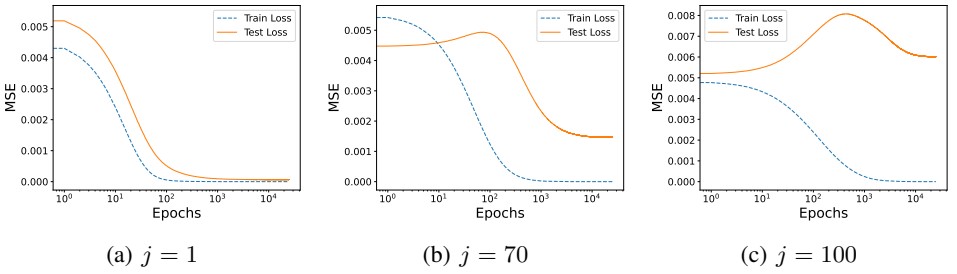

(a) $j = 1$      (b) $j = 70$      (c) $j = 100$

Figure 5: Demonstrating grokking on standard Gaussian $X \in \mathbb{R}^{D \times P}$ data with the one-hidden layer architecture we had for polynomial regression. Label vectors, $y$ are replaced with $j$-th largest eigenvectors of the initial NTK (ordered by descending eigenvalue) for $j \in \{1, 70, 100\}$, respectively for (a)-(c). If the network begins highly aligned as in (a), learning curves move together. If the network features are poorly aligned at initialization, as in (c), the network cannot generalize. In the middle, as in (b), the network groks.

**MNIST.** Liu et al. (2022a) induce grokking on MNIST by increasing parameter weight norm at initialization. In Figure 6(a) below, we make this grokking vanish continuously by tuning $\alpha$ without changing parameter weight norm, supporting the idea that weight norm at initialization is one of several parameters that control laziness. This tuning of $\alpha$ is done with the quadratic scaling demonstrated in Figure 7.

**One-layer Transformers.** In 6(b), we replicate the classic setting in which grokking was observed by Nanda et al. (2023), where a one-layer Transformer was trained on a modular addition task with cross-entropy loss and AdamW. We find that even in a vastly different network architecture – one with attention this time – an adaptive optimizer and different loss function, our hypothesis that laziness controls grokking holds out of the box. Note how changing alpha can continuously make grokking vanish, or more drastic.

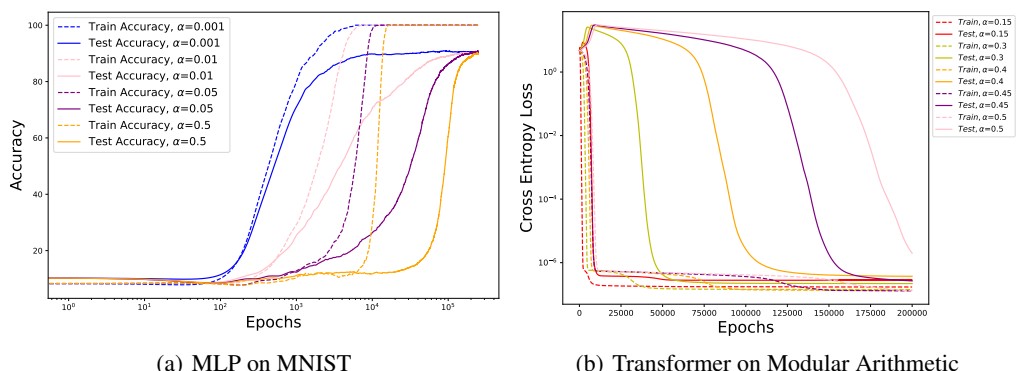

(a) MLP on MNIST        (b) Transformer on Modular Arithmetic

Figure 6: (a) **MNIST.** Continuously controlling grokking on MNIST by only varying laziness, $\alpha$, including making it vanish by encouraging fast feature learning at $\alpha = 0.0001$. (b) **One-layer Transformer.** Continuously controlling grokking on a one-layer Transformer from Nanda et al. (2023) using the scale parameter $\alpha$. Note how grokking has vanished for $\alpha = 0.15$, so it can barely be seen amidst the train curves. Axes scales are kept faithful to the original papers. We find the same trend with student-teacher networks, with plots in Appendix Section 8.

## 7 CONCLUSION

In this work, we hypothesized that grokking may arise from a transition from lazy to rich learning. To explore this idea, we sought the simplest possible model in which we can see grokking, namely polynomial regression. This motivated the examination of a laziness parameter, $\alpha$, and the initial NTK alignment, which we showed can induce and eliminate grokking out-of-the-box. While many prior works empirically found that parameter weight norm seemed an important factor for network grokking, we showed that this occurs because large initial weight norm approximately *linearizes* the model. In this way, our results subsume past work on grokking and show why weight decay is not always necessary.

**Limitations** While we provided a new conceptual framework for reasoning about grokking, some questions are left out of scope. We do not provide a sharp theoretical characterization of the required sample size or training time to transition from memorization to grokking behavior. While it is true that adaptive optimizers and weight decay are neither necessary nor sufficient for grokking to be seen, their ubiquity in past examples of grokking does suggest they can amplify grokking. One possibility is that weight decay encourages the model to leave the lazy regime by forcing changes to the NTK (Lewkowycz and Gur-Ari, 2020), which we explore in Section 13 of the Appendix. While we speculate on the role of the optimizers (momentum and Adam) in inducing grokking in Appendix 14, the dynamics at play when adaptive optimizers induce grokking remain poorly understood.

## ACKNOWLEDGEMENTS

TK was supported by the Harvard College Research Program. BB and CP are supported by NSF Award DMS-2134157. BB was additionally supported by a Google PhD fellowship. This work has been made possible in part by a gift from the Chan Zuckerberg Initiative Foundation to establish the Kempner Institute for the Study of Natural and Artificial Intelligence. We thank Andrey Gromov, Bruno Louriero, Francesca Mignacco, Stefano Sarao Mannelli, and Nikhil Vyas for useful discussions and thank Alex Atanasov for helpful comments on an earlier draft of this paper.

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

# 8 APPENDIX

ADDITIONAL EXPERIMENTS

## 8.1 A NOTE ON IMPLEMENTING LARGE $\alpha$

As was discussed in the original paper on inducing lazy training by Chizat et al. (2019), it is important to subtract off the initial predictor from the neural network function. Concretely, one should take as their predictor, the modified function

$$\tilde{f}(\boldsymbol{x}, \theta) = \alpha[f(\boldsymbol{x}, \theta) - f(\boldsymbol{x}, \theta_0)] \tag{7}$$

In our experiments, we place $\tilde{f}$ in the optimizer when we compute gradients. This allows us to train with very large values of $\alpha$ without experiencing instability at initialization. Further, to maintain consistent timescales of training, it is important to rescale learning rate as $\eta = \eta_0/\alpha^2$. This rescaling keeps the initial derivative $\frac{d}{dt} f|_{t=0}$ independent of $\alpha$ and thus the early training dynamics are consistent across $\alpha$. Otherwise, large $\alpha$ networks will have too large a learning rate and small $\alpha$ networks take too long to train.

## 8.2 WEIGHT NORM AND INITIALIZATION SCALE HAVE THE SAME EFFECT

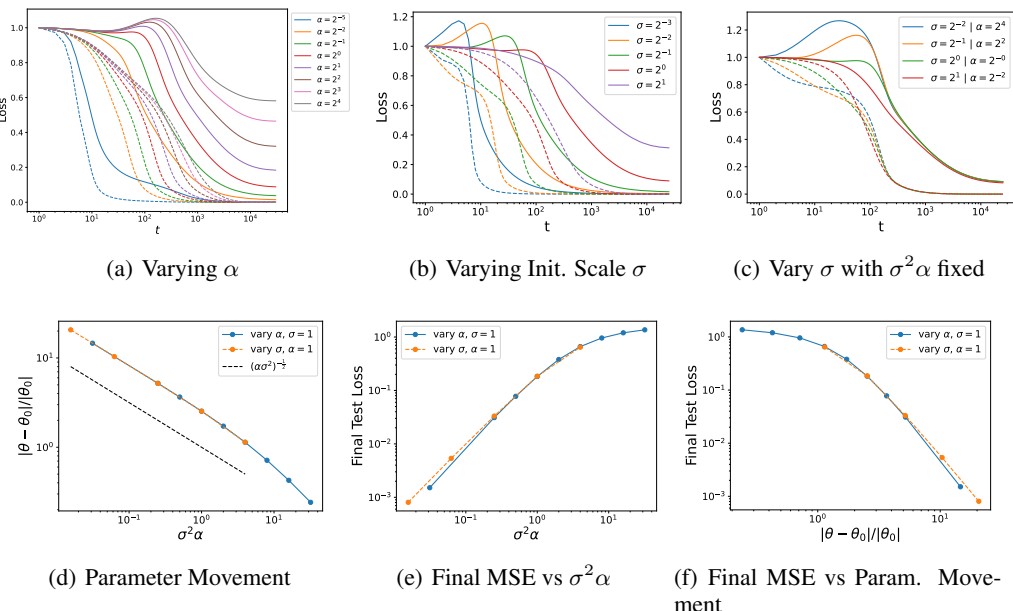

(a) Varying $\alpha$  (b) Varying Init. Scale $\sigma$  (c) Vary $\sigma$ with $\sigma^2\alpha$ fixed

(d) Parameter Movement  (e) Final MSE vs $\sigma^2\alpha$  (f) Final MSE vs Param. Movement

Figure 7: Weight scale alters grokking dynamics and final test loss primarily by manipulating the richness of the dynamics, specifically how far the dynamics deviates from the linear/kernel regime. (a) The loss curves for a two layer network trained on the quadratic polynomial with varying laziness parameter $\alpha$. Note that small $\alpha$ networks do not grok as dramatically as large $\alpha$ and also achieve good test loss. (b) A similar sweep over weight initialization scale $\sigma$ yields the same trend. (c) For a two layer network, approximate preservation of the feature learning can be achieved by choosing $\alpha \sim \sigma^{-2}$. (d)-(f) The *relative parameter change* $|\boldsymbol{w} - \boldsymbol{w}_0|/|\boldsymbol{w}_0|$ and final test loss is controlled through the quantity $\sigma^2\alpha$.

## 8.3 MODULAR ARITHMETIC IN A 2 LAYER PERCEPTRON

Here are the architecture details for grokking with weight norm increase on a modular arithmetic task on a two-layer perceptron with vanilla GD. First, we consider the classic modular arithmetic task $a + b \equiv c \mod p$ used previously to explore grokking in Power et al. (2022); Nanda et al. (2023). We find grokking in a one-hidden layer network with input dimension $D = 2p$, where the

input is a concatenation of $a, b$ in one-hot encoding, and the output is a dimension $p$ one-hot encoded vector $c \in \{0, 1, \cdots, p-1\}$. We use $N = 100$ as the hidden width, but a wide range of hidden widths exhibit grokking. We use $90\%$ of all $p^2$ possible pairs for training, and the rest of the test; the learning rate is $\eta = 100$. The figures below use $p = 23$, but we see the same curves for most integers $p$, for instance, any of $p \in \{13, 17, 19, 23, 29, 31 \cdots\}$ also give grokking in the same way. Crucially, we do *not use any weight decay*, and merely use *vanilla gradient descent*. Our results agree with those of Gromov (2023).

### 8.3.1 STUDENT-TEACHER NETWORKS

Consider a task where we initialize a "teacher" 5-100-100-5 MLP with $\tanh$ activations, and a student with the same architecture. The student is tasked with being trained to learn outputs generated by the teacher network on an input dataset $X$ of standard Gaussians. This task is introduced in Liu et al. (2022a) to show how grokking can be introduced by doubling the student's weight norm at initialization. Here, we make that grokking both vanish and more pronounced without touching their weight norm at initialization, showing our $\alpha$ parameter does the same thing because the deeper phenomenology of lazy training and model linearization is what is at play.

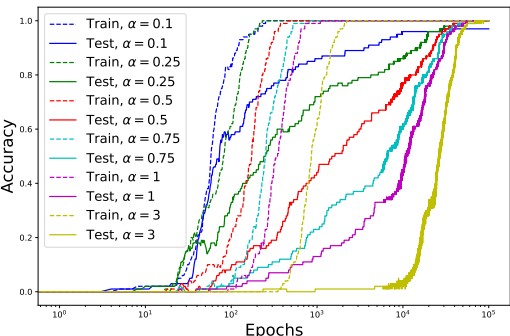

Figure 8: Controlling grokking on a student-teacher task defined in Liu et al. (2022a). The purple curve corresponds to exactly the experiment in their paper, grokking induced by increasing weight norm. See that by tuning our scaling parameter to be smaller $\alpha \to 0$, we can make that induced grokking vanish as on the left of the purple curve, and in fact, we can even make grokking more pronounced by increasing it. Since this is a regression task, accuracy is defined as the student being with 0.001 of the teacher label.

### 8.3.2 GOLDILOCKS DATASET SIZE

Here we see the three regimes of train set size: too little (a, d), just right (b, e), and too much (c, f). The top row is the learning curve, and the bottom row is the loss decomposition. Note that we can directly see the network putting power in linear components in (a). This linear power comes from the NTK at initialization, since the activation function has a linear term. This attempted (approximate) kernel method helps us interpolate the train set, but leads to *worse than random* behavior since the initial features are not matched to the task. Any deviations between train and test loss at initialization are because we use a data set size of $P = 50$. As we take data larger, the two match.

### 8.3.3 INTERPOLATING FROM RANDOM WEIGHTS TO ALIGNED SOLUTION TO MAKE GROKKING VANISH

We claim NTK alignment at initialization is one of two important factors that control grokking. Figure 1 suggests if we increase alignment at initialization, grokking should vanish. This happened as predicted in Figure 6 where we changed the labels, and now we'll tune alignment a different way: by changing the weights at initialization. Here, we start with random weights for our polynomial regression task, where we see grokking as usual, and gradually add a small component of the final solution (computed analytically) weights to the initial weights. This increases initial alignment, and

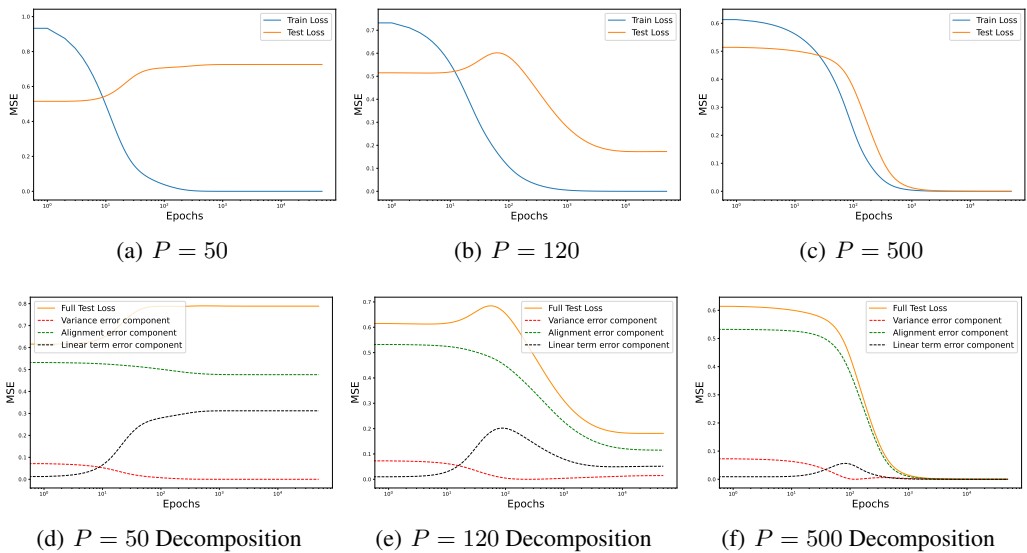

(a) $P = 50$     (b) $P = 120$     (c) $P = 500$

(d) $P = 50$ Decomposition     (e) $P = 120$ Decomposition     (f) $P = 500$ Decomposition

Figure 9: (First row) Goldilocks zone in dataset size. 2 layer perceptron trained on usual polynomial regression task with $P \in \{50, 120, 500\}$ in (a)-(c) respectively. (Second row) Loss decomposition for each of (a)-(c) respectively, showing how failure to generalize (a) comes from power in linear modes (black dashed), grokking comes from a rise then fall in power in linear modes (b) and full generalization comes from low power in linear modes throughout, and zero at end time (optimal features have no power in linear modes), as we see in (f).

of course, decreases initial loss a bit. The crucial observation in the plot below is that *increasing alignment by even a little bit makes dramatic grokking (increase in test loss alongside an initial decrease in test loss)* vanish. We see this as we go from the dark blue curve to the gold one.

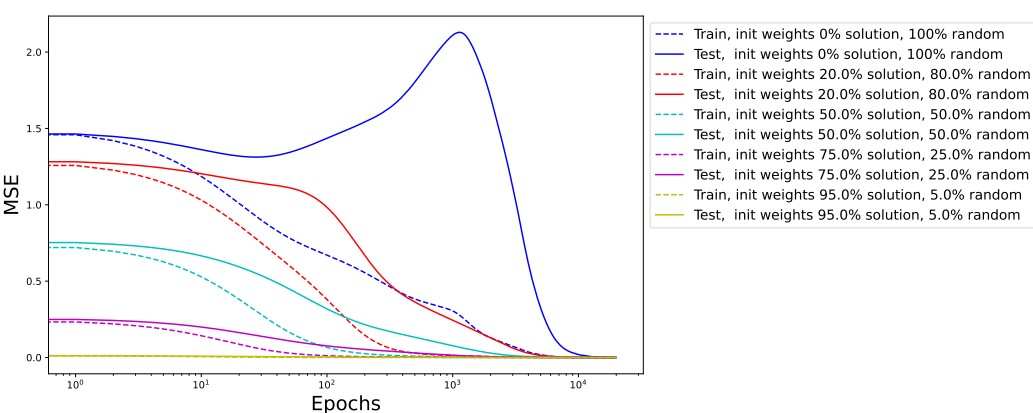

Figure 10: Varying alignment by adding some end-time solution to the initial weights. Grokking-like increase in test loss vanishes with even a small increase in alignment (solid blue to solid red) as we add a small component of the solution weights to the initialization weights. This is evidence that initial NTK alignment controls how much test loss suffers during the initial fall in train loss as more power is put into kernel regression solution.

## 9 RELATION TO OTHER THEORIES OF GROKKING

Existing grokking literature focuses on two things. One is studying modular arithmetic (and more generally group compositional) tasks and good representations to learn them, and the other is the empirical relationship between initial parameter weight norm, weight decay, and grokking.

Examples of papers exploring the first theme include the study of how a Transformer learns to do modular arithmetic in Nanda et al. (2023), Gromov (2023) writing out the closed-form solution of final trained weights for another modular arithmetic task, and Liu et al. (2022b) who find a circular embedding within the network for a modular arithmetic task. A similar result was found in the original Power et al. (2022) paper that discovered grokking. Examining the geometry of embeddings and learned representations like this is only possible when you *fix a specific task of interest*, and often these are tasks with a natural geometric structure. These papers often give compelling insights into what and how networks learn representations for certain specific classes of tasks, but Liu et al. (2022a) showed we can see grokking on other tasks by tuning initial weight norm, so while algorithmic tasks like modular arithmetic might have some interesting structure that makes grokking easier to induce, they are neither sufficient nor necessary for grokking to appear.

The two important papers exploring the second theme (parameter weight norm and weight decay) are by Liu et al. (2022a) as mentioned above, and Varma et al. (2023).

Liu et al. (2022a) induces grokking by increasing the initial weight norm and concludes generalizing solutions lie on smaller norm spheres in parameter space. We agree that initial weight norm can induce grokking, but we claim it is *not* because generalizing solutions always lie on smaller norm spheres in parameter space. Our modular arithmetic task in Section 2 and our polynomial regression task that is the focus of our paper are both counterexamples, since the final generalizing solution has a larger parameter weight norm than the initial solution, and we do not need weight decay to learn it or to grok. Instead, we claim initial weight norm controls grokking by linearizing the model (moving it into the "lazy training" regime). The different learning dynamics at large weight scale in Liu et al. (2022a) is quite reminiscent of the transition to lazy training in Chizat et al. (2019).

Varma et al. (2023) explain grokking as arising from three specific causes. Their claimed first and third causes correspond naturally to objects in our paper, and it is with their second cause that our theory (and experiments) disagree. Their first claim is that grokking requires "two families of circuits," one that memorizes and one that generalizes. This corresponds to our two regimes of lazy and rich training dynamics. Their third claim is that during grokking, the "memorizing circuit is stronger at early phases of training" which in our language corresponds to the network starting off lazy before learning features at late time.

Their second and core claim is where the two theories disagree. They claim that this transition from the memorizing to generalizing circuit happens *because the generalizing circuit is more "efficient" than the memorizing circuit, in the sense that it can produce equivalent loss with lower parameter norm*. In our paper, we showed two different tasks (modular arithmetic and polynomial regression) in which the generalizing solution that reaches zero loss is *higher parameter norm* than the memorizing solution the network started in. We do agree with their intuition that there are two *regimes* (lazy and rich) in which the network can operate, and parameter norm has something to do with which regime we are in. In fact, we explore the role of weight decay in Section 13, finding it consistent with our theory that weight decay can be helpful for inducing grokking.

*A note on data set size.* It has been long known that grokking happens in a particular data regime. In the paper introducing grokking, Power et al. (2022) mention in Section 3.1 of their paper that for large data set sizes, training and validation track each other, and Nanda et al. (2023) notes in Section 5.3 of their paper that with enough data, there is no longer a gap between train and test loss. Varma et al. (2023) study the behavior of learning curves around the critical dataset size $D_{\text{crit}}$, finding variants of grokking behavior. Note that being in the "goldilocks zone for data set size" is necessary but not sufficient to see grokking.

*Parities as a binary polynomial problem*: Barak et al. (2022) study learning of parities with a neural network which exhibits grokking-like dynamics when learned with two layer networks with SGD. The target function they consider has the form $y = \prod_{i \in S} x_i$ where $S \subseteq [n]$ is a subset of the first $n$ bits. The variables $x_i \sim \{\pm 1\}$ are random binary variables. For dot-product kernels, such as wide networks at initialization, this problem has the same sample complexity as learning polynomials

of degree $|S|$ with Gaussian data for kernel regression (Misiakiewicz, 2022). Our choice to study the learning dynamics of polynomial regression with Gaussian data is therefore closest to this prior work.

## 10 "Hard/Misaligned tasks" for the kernel cause a breakdown in linearization

We denote a "hard task" for the initial kernel as a task where the trainset label vector $\boldsymbol{y} \in \mathbb{R}^P$ and the kernel gram matrix $\boldsymbol{K} \in \mathbb{R}^{P \times P}$ satisfy

$$\text{Difficulty} \equiv \boldsymbol{y}^\top \boldsymbol{K}^{-1} \boldsymbol{y} \gg 1. \tag{8}$$

One way to motivate this quantity is to note that near the kernel limit, the amount that the parameters move during training under MSE is precisely given by this quantity

$$|\boldsymbol{w} - \boldsymbol{w}_0|^2 \sim \boldsymbol{y}^\top \boldsymbol{K}^{-1} \boldsymbol{y} \tag{9}$$

We see that the linearization approximation can break for difficult tasks since $|\boldsymbol{w} - \boldsymbol{w}_0|^2$ could be large (think about Taylor expansions around $\boldsymbol{w}_0$). Thus lazy learning on tasks of infinite difficulty is not self-consistent. All other things being equal, we expect to see more feature learning on tasks that are difficult for the initial kernel. This is especially true if the training set is sufficiently large.

## 11 Derivations for Toy Model

### 11.1 Initial NTK Diagonalization

At infinite width, the network has the following value under random initialization (since $\bar{\boldsymbol{w}} = 0$ and $\boldsymbol{M} = \boldsymbol{I}$)

$$K(\boldsymbol{x}, \boldsymbol{x}') = \boldsymbol{x} \cdot \boldsymbol{x}' + \epsilon^2 (\boldsymbol{x} \cdot \boldsymbol{x}')^2 \tag{10}$$

The Mercer eigenvalue problem for data distribution $p(\boldsymbol{x})$ has the form

$$\int d\boldsymbol{x} \, p(\boldsymbol{x}) K(\boldsymbol{x}, \boldsymbol{x}') \phi(\boldsymbol{x}) = \lambda \phi(\boldsymbol{x}') \tag{11}$$

We seek eigenfunctions $\phi(\boldsymbol{x})$ and eigenvalues $\lambda$ that satisfy the above equation for the Gaussian data density $\boldsymbol{x} \sim \mathcal{N}(0, D^{-1} \boldsymbol{I})$. We first note the following

$$\langle K(\boldsymbol{x}, \boldsymbol{x}') x_i \rangle = \langle (\boldsymbol{x} \cdot \boldsymbol{x}') x_i \rangle = \frac{1}{D} x_i' \tag{12}$$

which implies that $x_i$ is an eigenfunction with eigenvalue $\lambda_{\text{lin}} = \frac{1}{D}$. Next, we see that

$$\langle K(\boldsymbol{x}, \boldsymbol{x}') x_i^2 \rangle = \frac{2\epsilon^2}{D^2} x_i^2 + \frac{\epsilon^2}{D^2} |\boldsymbol{x}'|^2 \, , \, \langle K(\boldsymbol{x}, \boldsymbol{x}') |\boldsymbol{x}|^2 \rangle = \frac{\epsilon^2}{D} \left( 1 + \frac{2}{D} \right) |\boldsymbol{x}'|^2 \tag{13}$$

This implies the existence of $D$ eigenfunctions

$$\begin{aligned}
\langle K(\boldsymbol{x}, \boldsymbol{x}') [x_i^2 - c|\boldsymbol{x}|^2] \rangle &= \frac{2\epsilon^2}{D^2} (x_i')^2 + \frac{\epsilon^2}{D^2} |\boldsymbol{x}'|^2 - \frac{c\epsilon^2}{D} \left( 1 + \frac{2}{D} \right) |\boldsymbol{x}'|^2 \\
&= \frac{2\epsilon^2}{D^2} \left[ (x_i')^2 + \left( \frac{1}{2} - \frac{c}{2} (D+2) \right) |\boldsymbol{x}'|^2 \right]
\end{aligned} \tag{14}$$

The above is an eigenfunction if $2c = c(D+2) - 1$, or $c = \frac{1}{D}$. We have thus identified an additional $D$ eigenfunctions with eigenvalue $\lambda = \frac{2\epsilon^2}{D^2}$. Lastly, we consider $x_i x_j - c|\boldsymbol{x}|^2$ for $i \neq j$.

$$\langle K(\boldsymbol{x}, \boldsymbol{x}') x_i x_j \rangle = \frac{2\epsilon^2}{D^2} x_i' x_j' \tag{15}$$

which transparently gives us another $\frac{1}{2}(D^2 - D)$ eigenfunctions with the same eigenvalue $\frac{2\epsilon^2}{D^2}$. Lastly, we note that $|\boldsymbol{x}|^2$ is also an eigenfunction

$$\left\langle K(\boldsymbol{x}, \boldsymbol{x}')|\boldsymbol{x}|^2 \right\rangle = \frac{1}{D}\left(1 + \frac{2}{D}\right)|\boldsymbol{x}'|^2 \tag{16}$$

with eigenvalue $\frac{1}{D}(1 + \frac{2}{D})$. The target function has the decomposition

$$y(\boldsymbol{x}) = \frac{1}{2}\left[(\boldsymbol{x} \cdot \boldsymbol{\beta}_\star)^2 - \frac{1}{D}|\boldsymbol{x}|^2\right] + \frac{1}{2D}|\boldsymbol{x}|^2 \tag{17}$$

The first term in brackets has kernel eigenvalue $\frac{2\epsilon}{D^2}$ while the second term has eigenvalue $\frac{1}{D}(1 + \frac{2}{D})$. This suggests a kernel method would learn the function $\frac{1}{2D}|\boldsymbol{x}|^2$ at sample sizes $P \approx D$ but will only learn the full target function at $P \approx D^2$. This separation of timescales motivated the original investigation of this task in which to find grokking.

## 11.2 Kernel Alignment

For any configuration of weights, the NTK of our toy model has the following form

$$K(\boldsymbol{x}, \boldsymbol{x}') = \boldsymbol{x} \cdot \boldsymbol{x}' + \epsilon(\boldsymbol{x} \cdot \boldsymbol{x}')\bar{\boldsymbol{w}} \cdot (\boldsymbol{x} + \boldsymbol{x}') + \epsilon^2(\boldsymbol{x} \cdot \boldsymbol{x}')\boldsymbol{x}^\top \boldsymbol{M} \boldsymbol{x}'. \tag{18}$$

In this section, we compute the kernel-task alignment metric on the test distribution which is discussed in the paper. In particular, we will show that this is related to the correlation of $\boldsymbol{M}$ with $\boldsymbol{\beta}_\star \boldsymbol{\beta}_\star^\top$ which is also present in our misalignment error. This alignment requires computing

$$\frac{\langle y(\boldsymbol{x})K(\boldsymbol{x}, \boldsymbol{x}')y(\boldsymbol{x}')\rangle}{\sqrt{\langle K(\boldsymbol{x}, \boldsymbol{x}')^2\rangle \langle y(\boldsymbol{x})^2\rangle}} \tag{19}$$

The most important term is the numerator which has the form

$$\langle y(\boldsymbol{x})K(\boldsymbol{x}, \boldsymbol{x}')y(\boldsymbol{x}')\rangle = \frac{\epsilon^2}{D^2}\left[|\boldsymbol{\beta}_\star|^2 \left\langle (\boldsymbol{\beta}_\star \cdot \boldsymbol{x})^2 \boldsymbol{x}^\top \boldsymbol{M} \boldsymbol{x}\right\rangle + 2\left\langle \boldsymbol{\beta}_\star \boldsymbol{M} \boldsymbol{x}(\boldsymbol{\beta}_\star \cdot \boldsymbol{x})^3\right\rangle\right]$$

$$= \frac{\epsilon^2}{D^4}|\boldsymbol{\beta}_\star|^4 \mathrm{Tr}\boldsymbol{M} + \frac{8\epsilon^2}{D^4}\boldsymbol{\beta}_\star^\top \boldsymbol{M} \boldsymbol{\beta}_\star |\boldsymbol{\beta}_\star|^2 \tag{20}$$

We see that the numerator in the kernel alignment formula increases with the alignment of $\boldsymbol{M}$ to the $\boldsymbol{\beta}_\star \boldsymbol{\beta}_\star$ direction. This term also appears in the alignment error of the MSE decomposition for our problem.

## 12 Loss Decomposition With Readouts

If we instead train both the input weights $\{\boldsymbol{w}_i\}$ and readout weights $v_i$ in a two layer network

$$f(\boldsymbol{x}) = \frac{\alpha}{N}\sum_{i=1}^{N} v_i \phi(\boldsymbol{w}_i \cdot \boldsymbol{x}) = \alpha \bar{\boldsymbol{w}} \cdot \boldsymbol{x} + \frac{\alpha\epsilon}{2}\boldsymbol{x}^\top \boldsymbol{M} \boldsymbol{x} \tag{21}$$

where the new $\bar{\boldsymbol{w}}$ and $\boldsymbol{M}$ have the formulas

$$\boldsymbol{M} = \frac{1}{N}\sum_{i=1}^{N} v_i \boldsymbol{w}_i \boldsymbol{w}_i^T \;,\; \bar{\boldsymbol{w}} = \frac{1}{N}\sum_{i=1}^{N} v_i \boldsymbol{w}_i. \tag{22}$$

These $\bar{\boldsymbol{w}}$ and $\boldsymbol{M}$ can be plugged into the loss decomposition in the main text. At initialization, the kernel has the form

$$K(\boldsymbol{x}, \boldsymbol{x}') = 2\boldsymbol{x} \cdot \boldsymbol{x}' + \frac{3\epsilon^2}{2}(\boldsymbol{x} \cdot \boldsymbol{x}')^2 + \frac{\epsilon^2}{4}|\boldsymbol{x}|^2|\boldsymbol{x}'|^2 \tag{23}$$

This kernel is still strongly biased towards linear functions. As before, the eigenvalues associated with quadratic functions is $\mathcal{O}(\epsilon D^{-2})$.

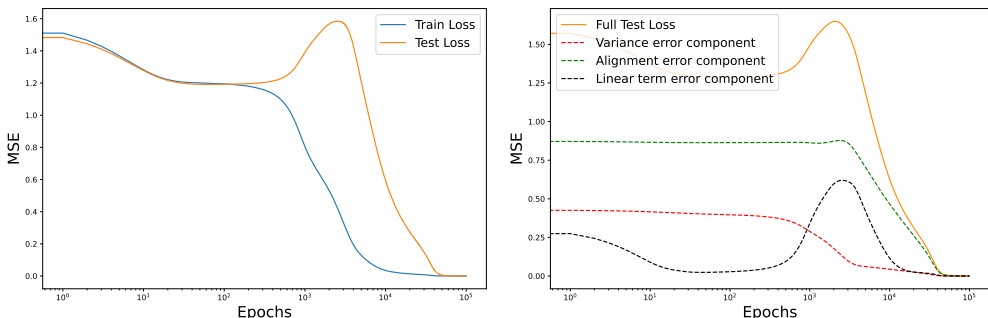

Figure 11: Grokking decomposition for an ordinary 2-layer MLP, ie. the committee machine in the main text with readout weights trained. The same mechanism holds as posited.

# 13 THE ROLE OF WEIGHT DECAY

While we showed that weight decay in a model is neither necessary nor sufficient to induce grokking, it is true it can be helpful in generating grokking. Here, we explore why that is, speculatively arguing that it is because weight decay moves us out of the lazy regime. In Lewkowycz and Gur-Ari (2020) it is proven that under weight decay, the NTK of a network continuously evolves over time. This means networks trained using large amounts of weight decay cannot stay lazy for long.

For some intuition, consider the following. Lazy training during gradient requires the network parameters to move only through a particular tangent space in a much larger parameter space. Informally, this is a somewhat fragile condition that requires some weights to move in a certain direction while others move in another direction to make sure the network NTK doesn't change. Weight decay is a penalty on weight norm *irrespective* of which weight is contributing to this norm, and so can be thought of as a perturbation that quickly moves us out of the lazy training regime. When a generalizing solution has a lower weight norm than our initialization, weight decay can encourage learning of useful features and thus generalization.

In Nanda et al. (2023), it is shown that the amount of weight decay can control the amount of grokking (time delay). This is reminiscent of how our laziness parameter $\alpha$ has the same effect. Indeed, we show here they have competing effects in this sense. In that paper, it is shown how the grokking time delay between train and test loss vanishes if weight decay is high enough. Our hypothesis would predict making the network lazier compensates, recovering the amount of grokking we started with. We see this happen below.

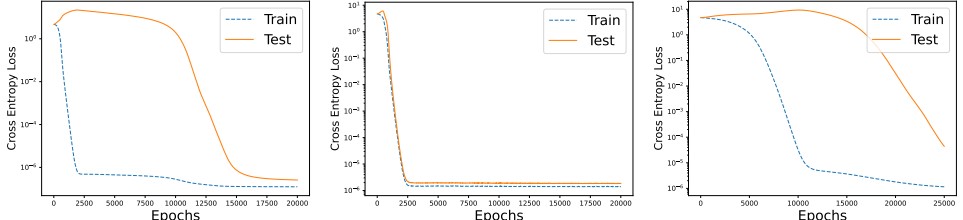

Figure 12: (a) Start with large grokking time delay with $\alpha = 0.5, \mathrm{wd} = 1$. (b) Increasing weight decay to $\mathrm{wd} = 5$ (keep $\alpha = 0.5$) makes grokking vanish by encouraging immediate movement in the NTK (feature learning). (c) Then commensurately increasing laziness to $\alpha = 1.4$ (keep $\mathrm{wd} = 5$) recovers grokking by discouraging feature learning. We believe past works see grokking with a high initial weight norm and weight decay because these two components imply initial laziness (high initial weight norm) and eventual feature learning (weight decay, as illustrated here).

One thing worth highlighting is that this explains why Liu et al. (2022a) found that time to generalize in the grokking setting there was linear in the amount of weight decay, $\kappa$. Moreover, Nanda et al. (2023) finds that the time taken to generalize drops by a factor of ten in Figure 27 of that paper,

when weight decay is increased by a factor of again, again linear. It turns out that Lewkowycz and Gur-Ari (2020) proves that the NTK evolves at order $O(\kappa)$ for $\kappa$ the weight decay parameter, *so our theory that grokking can be seen as the transition from the NTK to feature learning regime exactly predicts the quantitative effect weight decay has on grokking timescales!*

## 14 THE ROLE OF MOMENTUM AND ADAPTIVE OPTIMIZERS

This section is more speculative. In the main text, our focus is on networks that use vanilla gradient descent because they are sufficient to get networks to grok and a simple setting that can be analyzed. This was motivated in part by the fact that the dynamics of networks optimized with adaptive optimizers are not well understood theoretically. But given that almost every paper on grokking in the past has used AdamW (Power et al., 2022; Nanda et al., 2023; Liu et al., 2022a;b; Varma et al., 2023), it is worth some comment.

In (a) we repeat our polynomial regression setup and induce grokking as in 1, but now using momentum $= 0.95$ in the vanilla GD optimizer. It has broadly the same shape, and certainly the same pattern (linear then feature learning) but with more bumps in the learning curves, presumably reflecting the momentum in gradient descent. In (b) we want to point out another set of learning curves that generate grokking-like accuracy curves. These are the *loss* curves for the student-teacher task in Section 8 for which the *accuracy* curves we plotted showed stark grokking. That task uses AdamW as well, again showing the linearization tracking the model until the loss halves, before diverging then. The nonmonotonicity in late-time is presumably a product of the AdamW optimizer, much like such nonmonotonicity emerged in (a) from momentum.

Note that both (a) and (b) are characterized as grokking even though the loss curves look different: the key is delayed generalization. This shows how power in the initial kernel method is often unhelpful for generalization (a), but can sometimes be somewhat helpful in feature learning downstream, for instance in (b). Finally, (c) shows early-time linearization dynamics in the early stages of grokking with a one-layer Transformer from Nanda et al. (2023), which was trained using AdamW as well. Note how due to weight decay, the network loss actually quickly deviates from its linearization, despite the presence of grokking. This, along with the fact that scaling $\alpha$ still tunes grokking, is evidence that the transition from lazy to rich dynamics is a sufficient but not necessary mechanism underlying delayed feature learning on general tasks and settings. If we look at Figure 7 of that paper, we see this early time deviation coincides with a large early time increase in initial parameter weight norm, which could be due early-time behavior of adaptive optimizers Lewkowycz and Gur-Ari (2020); Cohen et al. (2021); Thilak et al. (2022), hence then claim that dynamics induced by adaptive optimizers in settings that grok are still not fully understood.

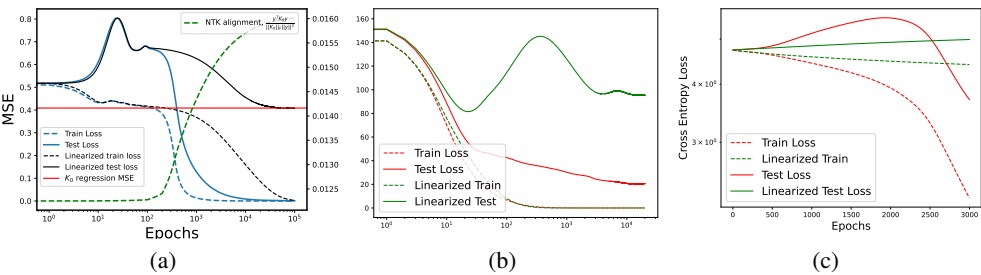

Figure 13: Learning curves for grokking with momentum and adaptive optimizers.

## 15 VARIANTS ON THE POLYNOMIAL TASK

The single-index quadratic learning task is known to have well-understood structure from a theory perspective Arous et al. (2021); Nichani et al. (2022); Bietti et al. (2022). A natural question is whether the grokking behavior and interpretation persists under ablation of either of these conditions. In particular, we consider the harder tasks of learning multi-index models, and higher degree (Hermite) polynomials. We find that grokking persists out-of-the-box as expected.

## 15.1 LEARNING MULTI-INDEX MODELS

It is known Nichani et al. (2022) that single index models can exhibit a transition from lazy to rich training dynamics, but in general less work has been done on multi-index models Arous et al. (2021). Thus, one might wonder whether grokking persists as we add multiple directions to the target function. That is, can we recover grokking with a target resembling $(\beta_1 \cdot x + \beta_2 \cdot x)$ (double-index model), or even $(\beta_1 \cdot x + \beta_2 \cdot x + \beta_3 cdotx)$ (triple-index). It turns out grokking does indeed persist in our model even if the target function spans multiple directions, as we see below.

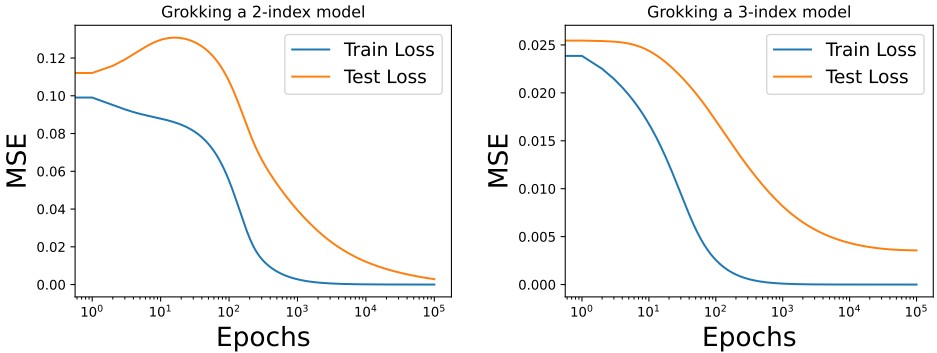

Figure 14: Grokking in our set-up persists when the target function is a higher-degree polynomial (with activation modified accordingly), so does not rely on special properties of the information exponent $k = 2$.

## 15.2 LEARNING HIGHER DEGREE POLYNOMIALS

For the task of learning a single-index model, it is known Arous et al. (2021); Nichani et al. (2022) that sample complexity and learning dynamics depend heavily on a quantity of target function called the *information exponent*. For simple polynomial target functions, as in our setting, this is simply the degree of the polynomial. One salient result is the qualitative difficulty of learning a quadratic target (where the information exponent is $k = 2$), and higher-degree targets (where the information exponent is $k \geq 3$). For instance, Theorem 1.3 in Arous et al. (2021) illustrates the difference in learnability and sample complexity for these cases. Thus, a natural question is whether grokking arises from special properties of a quadratic target here. The answer to this turns out to be no: the grokking examined in this paper (that is: polynomial regression in a 2 layer MLP with zero weight decay and vanilla GD) persists out-of-the-box for higher degree polynomial targets in the way one would expect. Below, we show experiments where the target is $H_k$, the $k$-th Hermite polynomial, and the activation is $\phi(z) = z + \frac{\epsilon}{2} H_k(z)$. In particular, $H_3, H_4$ below are polynomials with degree $k > 2$ and thus information exponent also greater than two. We use larger amounts of data to learn higher order polynomials $H_k$.

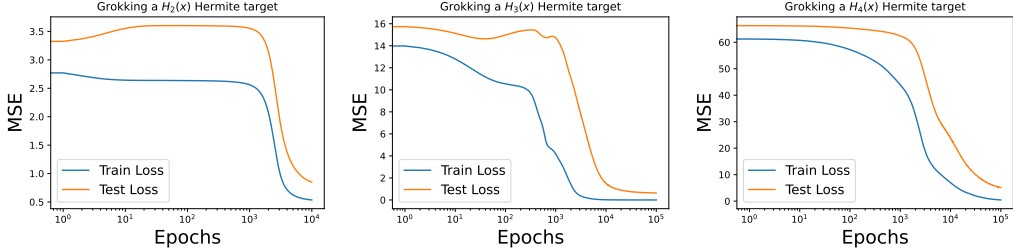

Figure 15: Grokking when learning the Hermite polynomials of varying degree illustrates that our setup doesn't rely on any special properties of a single-index quadratic learning problem, which is known to be theoretically simpler than higher degree or multi-index learning problems. Note the log-scale on $x$-axis, so a small gap on the right of the plot corresponds to generalization delayed (grokking) by several thousand epochs.

## 16    STUDYING LOSS VS ACCURACY IN GROKKING LEARNING CURVES

### 16.1    DEFINING "GROKKING IN LOSS" AND "GROKKING IN ACCURACY"

"Grokking in loss" refers to the loss curves we study in this paper, where the train loss initially decreases while the test loss is nondecreasing, with the test loss eventually falling as the network generalizes. This involves the train and test loss both moving a nonzero amount initially (usually it means an initial period where the test loss rises as the network memorizes the train set) and usually involves nonzero (but low) test loss at end-time of training. An example of grokking in loss is to the right of the Figure below.

"Grokking in accuracy" refers to a sharp increase in test accuracy long after a commensurate increase in train accuracy. In particular, test loss stays flat at the beginning, and rises to perfect at late-time. An example of grokking in accuracy is given on the left of the Figure below.

Given that these two definitions seem somewhat different on first glance, and that the second is the "original" definition of grokking Power et al. (2022) on the task (modular arithmetic) on which it was discovered, one might wonder whether the definition studied in this paper (grokking in loss) is the same as that studied historically in past literature.

The answer to this question is yes. The two curves on the bottom correspond to the same network run. While Power et al. (2022) present the phenomenon in terms of accuracy (like the plot below, left), really the network is trained on loss (below, right). In particular, we notice two things:

- When we plot accuracy, we see the network does perfectly in terms of classification accuracy on the modular arithmetic task.
- However, the loss at end-time is nonzero. In this case, this does not contradict the above because the network is trained on one-hot inputs, and so if the maximum element in the network output is the correct index of the modular addition problem, the input will be correctly answered (accuracy) but will incur loss until the network put probability one on the index of the correct answer.

We also notice that

- Test accuracy is flat early in training, but test loss moves.
- Train accuracy reaches perfect classification in 2k epochs, when train loss is about $\approx 0.06$, but train loss keeps falling much after that all the way to zero.

The intuition for the first bullet is that during memorization, test loss decreases because the network is attempting to fit a linearized (NTK) solution at early time. Of course, since the network has not learned the structure of the task by this point (before 2k epochs), this is not helpful on test points. But we notice that this linearized solution is enough to do perfectly on the train set! If a

loss of $\approx 0.06$ corresponds to perfect classification, then we see that the test loss only dips below this amount around 3-4k epochs, *precisely when the network generalizes in accuracy.* Finally, the accuracy curve is flat at early time because we cannot do worse than zero accuracy, so the unhelpful change in weights that allow the network to interpolate the train set are counterproductive for the test set (as we see in terms of loss), but the accuracy is already zero, so the test accuracy curve stays flat. This is why the same network on the same run of the same task having an accuracy curve on the left derived from the loss curve on the right, is perfectly consistent. The two definitions of grokking are equivalent.

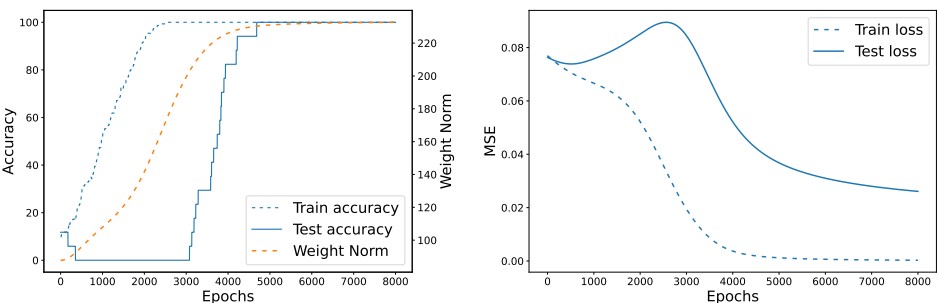

Figure 16: Accuracy and loss plots for the same network run of a modular arithmetic task. This type of task has historically been of interest in grokking, and is also studied in Power et al. (2022); Nanda et al. (2023); Gromov (2023). These plots are for a two layer network with no weight decay doing addition modulo a small prime.

Further evidence for the equivalence between the two measures can be seen by revisiting the Appendix of the original paper that introduced grokking Power et al. (2022), and noting Figure 4 which exhibits grokking in loss of the sort we study in this paper.

## 16.2 CONSTRUCTING ACCURACY CURVES FROM LOSS CURVES CAN BE MISLEADING

Some classes are classification tasks, such as modular arithmetic (as above) and classifying images on MNIST. There is a natural notion of accuracy in such tasks. However, regression tasks can also have a notion of classification accuracy defined, as is done Liu et al. (2022a), by defining a threshold $\theta$ such that a data point $(x, y)$ is said to be "classified correctly" if $|f(x) - y| \leq \theta$ it is within a threshold of the true label. Then one can define accuracy on a test set $X_{\text{test}}$ as the average number of data points in the set that are classified correctly $\frac{1}{|X_{\text{test}}|} \sum_{x_i \in X_{\text{test}}} \mathbf{1}[|f_{\text{net}}(x_i) - y_i| \leq \theta]$. The plot below does exactly this for the student-teacher task described earlier, taken from Liu et al. (2022a), showcasing how the resulting accuracy curves can be misleading and unhelpful, showing apparent "grokking" for some choices of $\theta$ but not others.

## 16.3 NOT ALL INSTANCES OF GROKKING IN ACCURACY ARE WORTH STUDYING

Reconsider the student-teacher task from above for which we saw that grokking in terms of accuracy could be an artefact of choosing a threshold $\theta$. We now examine its loss curves (below, left) and see they are entirely unremarkable. Thus, seeing a network "groks" in terms of an accuracy plot does *not* imply interesting learning dynamics in terms of loss (which is what guides learning dynamics). Therefore, grokking in accuracy does *not* imply grokking in loss. The curves below say more about the contrived nature of this task than about interesting learning dynamics in the network.

## 16.4 LOSS IS THE RIGHT METRIC TO STUDY

We have established that grokking in terms of loss curves, as we defined it in this paper, *does imply grokking in accuracy curves, as defined in Power et al. (2022).* We have also shown the converse is *not* true, demonstrating that grokking in loss, the way we have defined it, is a *stronger* condition than grokking in accuracy. This justifies our study of learning curves in terms of loss, not accuracy.

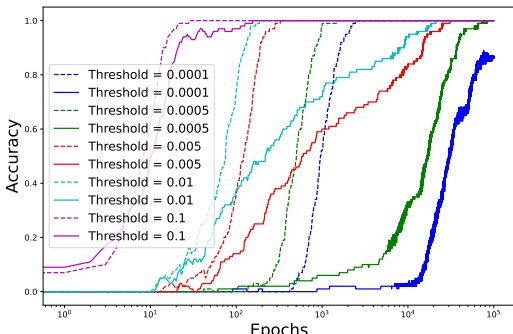

Figure 17: Grokking on this particular task depends on the choice of threshold. Each of these curves corresponds to the *exact same network run* (in terms of loss over epochs) with a different choice of threshold to plot accuracy. Figure 9 had the same threshold throughout. This justifies our study of loss, not accuracy, on regression-based tasks like polynomial regression in our main text.

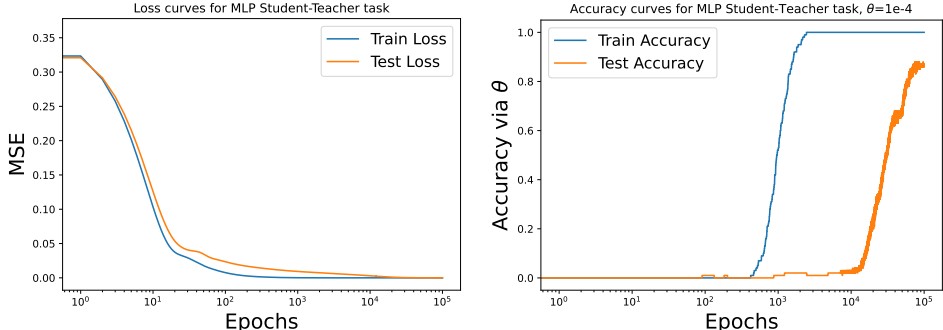

Figure 18: *Loss* curves for the student-teacher task on which *accuracy* curves exhibited "grokking." The loss curves show that there is nothing particularly surprising happening in the network over the course of training; the train loss tracks the test loss throughout. "Grokking" on the corresponding accuracy curves is thus a mirage arising from sweeping over thresholds $\theta$ used to define "accuracy" on this regression task, and choosing a $\theta$ that generates a striking plot.

Of course, as we saw in the case of the MLP on MNIST, modular arithmetic tasks, and more, this means that our claims for how laziness and NTK alignment control grokking (in terms of loss) hold *out of the box* for grokking (in terms of classification accuracy).

We end by noting that *we are not the first to make this observation.* Indeed, the fact that loss, not accuracy, is the interesting thing to study was quickly realized after the initial paper discovering grokking was published! It is noted when Davies et al. (2023) study loss in their Figure 5b, when Liu et al. (2022a) characterize the *loss* landscapes of various tasks that exhibit grokking, when Nanda et al. (2023) plots excluded and restricted loss and remarks that accuracy can be gamed, and emphatically in Schaeffer et al. (2023), who argues "hard threshold" measures of performance (like accuracy in grokking) are extremely misleading, and continuously optimized measures (like loss) should be studied instead. Thus our choice to do so in this paper is perfectly consistent with past literature on grokking.

## 17    MORE DATA ALLOW US TO GENERALIZE ON HARDER TASKS

If grokking is roughly seen as "delayed generalization," then one would expect we cannot grok on tasks where we cannot generalize. This is indeed the case, and one might wonder whether increasing the dataset size allows us to generalize on harder tasks – in particular on our eigenvector label task as in Figure 5 – or whether this task is inherently hard for a feature learning network. We will see below that more data allows us to generalize on this task when we otherwise couldn't. Reconsider the task in Figure 5(c) of learning the $j$-th eigenvector of Gaussian dataset $X$, for varying values of $j$. We chose this task because it allows us to vary task difficulty as measured by the alignment at initialization of the NTK and task labels. We observe that for any fixed $j$, increasing the dataset size suffices to eliminate grokking by having the learning curves move together. Consider the grid of plots below. It shows how for a fixed dataset size, making the task harder (increasing $j$) can cause a network to fail (going down the left column). But then we notice (moving left to right on each row) that increasing the dataset size leads to eventual generalization, with grokking in a goldilocks somewhere in the middle, when the task is hard (in an NTK alignment sense), but ultimately learnable (in the sense that the network can achieve low test loss at end-time of training).

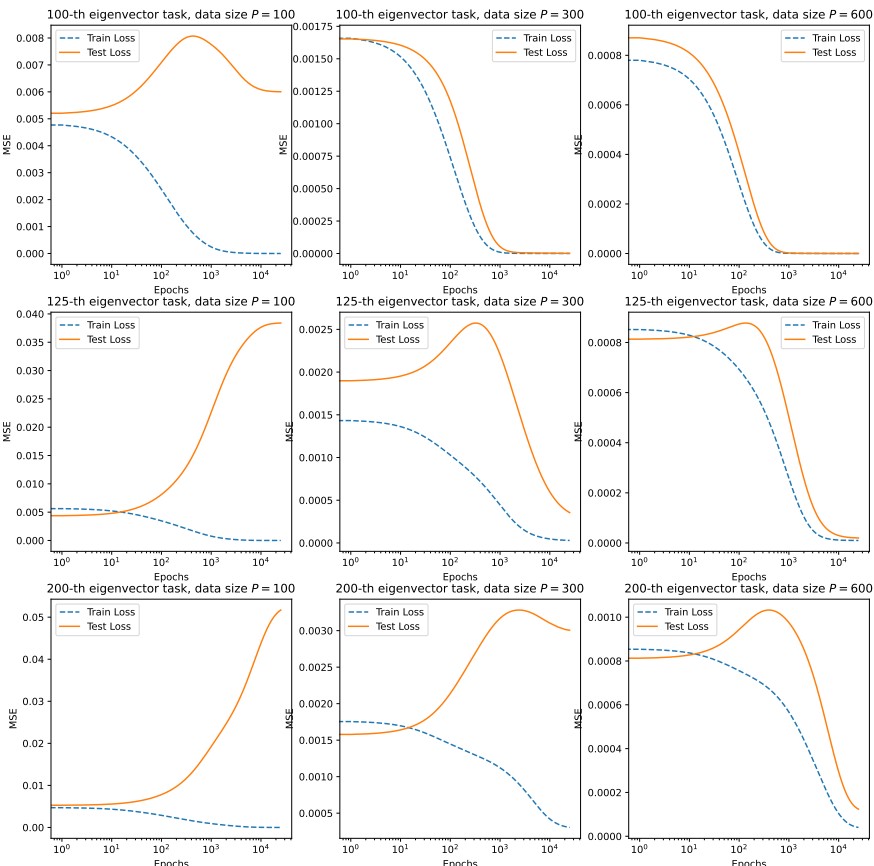

Figure 19: Varying dataset size and task difficulty to see where rich learning is enough to generalize. Plots going from left to right in each row have increasing amounts of data, and plots going from top to bottom have the task getting harder (increasing $j$ used for labels). We see that moving left to right illustrates (1), how increasing data for a fixed task difficult $j$, suffices to allow the network to generalize, but moving top to bottom on the grid illustrates (2), how if you let the task difficulty increase with dataset size, some tasks cannot be learned by this network architecture: if we used even smaller eigenvectors (for instance, setting $P = j$), no amount of $P$ would be able to learn it.

