# OpenReview forum: "Grokking as the transition from lazy to rich training dynamics"
_ICLR.cc/2024/Conference — ICLR 2024 poster_

### Official Review · Reviewer_8XT2 · 2023-10-17

**Soundness:** 3 good
**Presentation:** 4 excellent
**Contribution:** 3 good
**Rating:** 8
**Confidence:** 4

**Summary:**

The paper studies the grokking phenomenon and argues that the drop in test loss (after the drop in train loss) coincides with the network transitioning from the lazy to the rich regime. Grokking would then happen when the NTK at initialization is ill-aligned with the task, leading to a large test error in the lazy regime, however after the transition to the rich regime, the `right' features are learned.

They study a simple model of shallow nets that can fit linear and quadratic function, with a hyper-parameter that allows one to tune how much stronger the alignement of the initial NTK with the linear modes than quadratic ones. For a quadratic true function, one can ensure a bad alignement between the NTK and task, thus leading to grokking.

Finally this theory is tested on more practical examples (MNIST with a MLP and arithmetic with a transformer architecture).

**Strengths:**

There has been a lot of work studying grokking in different settings, but we are still lacking a complete understanding of under which condition it appears (model, training method, task). I think that the lazy/rich transition is the right idea to understand grokking.

Under this interpretation, we expect grokking to appear in tasks that are adapted to the rich regime but not the lazy regime. Furthermore, we need models that exhibit both rich and lazy regimes, and finally we need a training method where the model switches from lazy to rich regime in the middle of training.

The authors identify a simple theoretical model that exhibits grokking, and its simplicity allow for some theoretical results. The numerical experiments suggest that their interpretation applies to more realistic setups.

**Weaknesses:**

The theoretical and experimental evidence given in the paper is not particularly strong: in spite of the simplicity of the model considered, the theoretical results are very limited (just a decomposition of the loss into three terms with relevant interpretations); conversely most empricial results are done on this simple model, with just three experiments done on more complex data.

**Questions:**

You argue in a few different places that grokking when the NTK is ill-aligned to the task, but it feels to me that another condition is that the task needs to be well adapted to the rich regime (which I understand is much harder to characterize). There are tasks where neither the lazy nor the rich regime generalize, in which case one expects no grokking.

In section 6.1 you select a true function that is explicitely ill-aligned with the initial NTK to induce a grokking phenomenon, but the grokking phenomenon disappears when using too small eigenvalues. You argue that this is because there is not enough data, did you test this hypothesis? Could it also be the case because the eigenfunctions corresponding to small eigenvalues are hard to learn regardless of the regime (as I propose above)?

---

> ### Author Response · Authors · 2023-11-16
> **Response**
>
> We thank their reviewer for their comments and suggestions to improve the theoretical components of the paper. Below we discuss their questions and weaknesses and describe what we did to improve it.
>
> **Limited Theoretical Component to the Work**
>
> Asking for more theory was common amongst the reviewers, so, as outlined in the global response, we have included theory we had not included in the initial submission. Specifically, we derive limiting dynamics for quantities that determine the train and test losses when two layer networks are trained on high dimensional data. We show that the derived theory is accurate in Figures 4 and 13 and use it to derive approximate test loss and train loss gaps as functions of training time. Because these results are now so much stronger than the loss decomposition, we moved that to the Appendix.
>
> **Can Grokking Happen if Rich Regime Can't Generalize?**
>
> You are right that since grokking is roughly "delayed generalization," the network must be able to generalize in the first place from this sized training set. You make a good point here that some tasks may not even be soluble by a given network architecture in the rich regime no matter how much data you give it, because, for instance, the network architecture may not be expressive enough or have a high enough model capacity. To be clear, you are right that we assume the network is capable of generalizing on the task to begin with to be able to see grokking.
>
> **NTK Alignment and Data Quantity**
>
> This is an excellent question; we have added a new Section 17 to the Appendix to test this explicitly. We show that indeed even for task labels set to small eigenvectors, large amounts of data suffices to learn them and generalize. However, if you let the task become a *function of the dataset size* by defining the labels to be the smallest eigenvector of the empirical kernel matrix (in eigenvalue), then the task gets commensurately harder as you increase dataset size, and so even feature learning cannot allow this architecture to learn the task. This corresponds to going down the columns of our new Figure 21 and seeing that if we set $j = P$ then no amount of data will allow this architecture to generalize perfectly, giving no grokking.

---

> > ### Author Response · Authors · 2023-11-21
> >
> > Please let us know if there are any remaining questions before the end of the discussion period and if the added experiments and theory help to address the original concerns. Thank you for your time.

---

### Official Review · Reviewer_sTxd · 2023-10-23

**Soundness:** 3 good
**Presentation:** 3 good
**Contribution:** 3 good
**Rating:** 8
**Confidence:** 3

**Summary:**

This work hypothesizes that grokking may happen as a transition from lazy to feature learning regime. It also shows that grokking is not inherently associated with an increasing norm and the use of weight decay -- which advances our understanding.

**Strengths:**

I think the paper is clean and the point being made is sufficiently important. Also, I find the experiments sufficiently convincing.

Crucially, it advances our understanding of the grokking phenomenon. The shortness of the review only reflects the fact that I do not much much criticism for this work.

**Weaknesses:**

Nothing major.

**Questions:**

One point I think worth being clarified is the role of weight decay. I think we have a pretty good understanding of how weight decay changes the kernel. Essentially, weight decay almost always changes the NTK at the speed of $\kappa$, which is the strength of weight decay. The time scale for the effect of weight decay to take place is nothing but $1/\kappa$. Therefore, if $1/\kappa$ is far larger than the time scale of lazy training, one has a separation of time scale, where lazy training happens first, and feature learning induced by weight decay happens later. I think the authors should make more discussion on this point as it constructively adds to the paper.



------

This part is updated to acknowledge that I have read the author response.

---

> ### Author Response · Authors · 2023-11-16
> **Response**
>
> We thank the reviewer for their supportive comments and careful reading. Below we attempt to address their question about weight decay.
>
> **Role of Weight Decay**
>
> You are absolutely right. As you point out, papers like (Lewkowycz et al, 2020) show that the NTK evolves linearly as a function of weight decay. In response to your review, we have reworked Section 14 of the Appendix to explicitly discuss the role of weight decay, finding that past observations (such as those by Nanda et al, 2023, and Liu et al, 2022b) that empirically find weight decay has an approximately linear relationship with time to generalize (grok), our theory of grokking as a transition from lazy to rich dynamics predicts this exactly because, as you point out, the NTK evolves at $O(\kappa)$, so that the network is forced to leave the lazy (static NTK) regime at $t \sim O(\kappa)$ as well, which is what has been observed in past papers like Liu et al 2022b, but without explanation. Section 14 also illustrates how the laziness parameter $\alpha$ can supplant and replace weight decay in this way: making a network richer by decreasing $\alpha$ can serve the same effect as forcing NTK evolution (feature learning) by increasing weight decay, $\kappa$, and we showcase an example of that trade-off in Figure 14.

---

> > ### Author Response · Authors · 2023-11-21
> >
> > Please let us know if there are any remaining questions before the end of the discussion period. Thank you for your time.

---

### Official Review · Reviewer_y5Qp · 2023-11-05

**Soundness:** 2 fair
**Presentation:** 2 fair
**Contribution:** 2 fair
**Rating:** 3
**Confidence:** 3

**Summary:**

The paper proposes that the "grokking" phenomenon—where a neural network's test loss decreases significantly after the training loss—is due to a shift from initial "lazy" training to subsequent "rich" feature learning. Through a polynomial regression study with a two-layer network, the authors show that grokking occurs when the network moves from fitting data with its initial features to learning new features for better generalization. They suggest that the rate of feature learning and the initial feature alignment are key to this delayed generalization, a concept potentially applicable to more complex neural networks.

**Strengths:**

This paper studies the grokking phenomenon in deep learning, which is a recent hot topic and very relevant to ICLR. This paper proposes that grokking can be triggered by the transition from kernel regime to feature learning regime. Though this is already known even before the grokking paper by Power et al, 2022, e.g., the example of quadratically overparametrized linear model in Section 6 of Li et al., 2021, the novelty here is that this paper focuses on vanilla GD. In contrast, the transition between kernel and feature learning regime, Li et al., 2021, is triggered by sharpness minimization implicit bias of label noise SGD, which makes the analysis significantly easier.  This paper also performed some new experiments on various tasks to support this theoretical intuition.

*Reference:*

Li, Zhiyuan, Tianhao Wang, and Sanjeev Arora. "What Happens after SGD Reaches Zero Loss?--A Mathematical Framework." arXiv preprint arXiv:2110.06914 (2021).

**Weaknesses:**

1. The definition of grokking seems to be very different than that in literatue. In Power et al., 2022, **Groking** refers to the phenomena that "long after severely overfitting, validation accuracy sometimes suddenly begins to increase from chance level toward perfect generalization". While this paper describes grokking in their introduction as "train loss of a neural network decreases much earlier than its test loss".

	It is ok to me that this paper only focuses on the regression setting and talks about MSE loss. But the original grokking phenomenon is much more "suprising" and rare in the sense that it requires the test loss/error to get higher than random guess only after reaching almost zero training loss/error. Moreover, the original definition of grokking requires almost perfect generalization in the end of training.

2. The theoretical explanation presented in this paper only works for the new definition of grokking, but not the original definition of grokking in Power et al., 21. For the most experiments in this paper where the authors claim grokking happens, two criterions of original definition of grokking are not met: 1. when test loss starts to decrease before training loss gets small; 2. the eventual test loss is still high. Figure 5(b) is an example.

3. There is no end-to-end theorem about grokking. Given the modified definition of grokking, explanation using transition from kernel regime to feature learning regime seems quite straight forward. It should be very standard to show that there exists a period of time where training loss can decrease but test loss has a lower bound due to live in the kernel regime. Still this paper lacks such precise and rigorous statement.

4. I find it difficult to understand the math derivation for the toy model in section 11. In the paragraph above section 11.2, the authors write that kernel method can learn $|x|^2$ at D sample sizes and the full target function at $D^2$ sample sizes. How is this gap in sample complexity related to the sepration of timescale in a single run?

5. The first paragraph of Introduction does not sound correct. For example, the authors write that "Typically, one assumes that a model’s performance on its training set will be a reliable indicator of its generalization capabilities on unseen data". This is in general not true, especially in deep learning where models are often overparametrized. And it is exactly because training loss and test loss can have a large gap in practice, people are studying generalization in machine learning. Also it is not clear to me that why using models which groks in training will cause any safety issue. I would like the authors to elaborate on this or at least provide an example. Intuitively, grokking as a training behavior has nothing to do with the performance of the model when used for inference.

6. As mentioned in the strengths, the explanation of grokking using transition from kernel to feature learning regime (Li et al., 2021) is known even before the original grokking paper (Power et al., 2022). Though this paper is novel in the sense that it focuses on vanilla GD, which is more natural than Li et al., 2021, the theoretical result in this paper is also not as clean as that in Li et al., 2021.

*Reference:*

Li, Zhiyuan, Tianhao Wang, and Sanjeev Arora. "What Happens after SGD Reaches Zero Loss?--A Mathematical Framework." arXiv preprint arXiv:2110.06914 (2021).

**Questions:**

See weakness

---

> ### Author Response · Authors · 2023-11-16
> **Response Part 1**
>
> We thank the reviewer very much for their detailed review. Before we address the questions posed in the weaknesses section, we wanted to address some comments in the "Strengths" to make sure we are on the same page about the grokking literature and how this paper situates itself therein.
>
>
> *Strengths:
> ...This paper proposes that grokking can be triggered by the transition from kernel regime to feature learning regime. Though this is already known even before the grokking paper by Power et al, 2022, e.g., the example of quadratically overparametrized linear model in Section 6 of Li et al., 2021, the novelty here is that this paper focuses on vanilla GD. In contrast, the transition between kernel and feature learning regime, Li et al., 2021, is triggered by sharpness minimization implicit bias of label noise SGD...*
>
> First, you are correct that the main proposition here is that grokking can be seen as the transition from a kernel to feature learning regime. And to be clear, we are not claiming to be the first to show a network can transition from a lazy to rich training regime. Our novel contribution is to use this transition as a possible explanation of the grokking phenomenon, which to our knowledge has not been done before.
>
>
> 1. First, we thank you for bringing the paper by Li et al (2021) to our attention; we have cited it as it is relevant. However, it is a paper about the implicit biases of SGD around local minima, and how SGD can trigger a transition from lazy to rich training dynamics. However, SGD is *neither necessary nor sufficient to see grokking*, so no complete theory of grokking can be one entirely about SGD dynamics.
> 2. None of the existing papers on grokking mention the notion of NTK or lazy and rich training paradigms (including [(Liu et al, 2022a), (Nanda et al, 2023), (Liu et al, 2022b), (Davies et al, 2023), (Thilak et al, 2022)]). It is true that many papers remark something along the lines of "feature learning is somehow important" and "there is something resembling a phase transition happening" but the key contribution of this paper is *precisely in formalizing that claim in a way that allows us to demonstrably control and manipulate grokking in a way nobody has been able to do before.*
> 3. Many of the current popular theories of grokking claim that it is fundamentally due to weight decay [(Nanda et al, 2023), (Varma et al, 2023)] or weight norm [(Liu et al, 2022b)] or adaptive optimizers [(Thilak et al, 2022)], or any number of other things *that are directly inconsistent with the fact that grokking arises when a network moves out of the NTK regime* (as we discuss more in Section 2 and Section 10 of our paper). We think our paper can help illustrate how mechanically lazy/rich training regimes can alter the grokking curve.

---

> ### Author Response · Authors · 2023-11-16
> **Response Part 2**
>
> **Difference Between Accuracy and Loss Curves**
>
> The difference between accuracy and loss curves is important to clarify, so thank you for asking this.
>
> First, we added a two-page Section 17 to the Appendix explicitly addressing the questions here, including four new experiments to support our claim that loss is the right primitive to study. But we begin by first addressing your main concern that accuracy based curves in (Power et al, 2022) are more surprising than the loss curves we study in this paper because of the two criteria you give: *(Power et al, 2022) give the loss curves corresponding to their striking accuracy curves (Fig 1) in the appendix of their paper, as Figure 4. They look like the loss curves we study in our paper here*! The loss in our main toy model of study (Fig 1, Fig 3, Fig 11, Fig 12) looks exactly the same, and even in other experiments we do with more complicated data (Fig 6b, for instance), our curves also look like that. As far as your citation of Fig (5b) is concerned, we created a new Appendix Section 17 to show how curves that reach nonzero (but low) end-time loss give rise to accuracy curves that exhibit striking grokking. Note also that (Gromov, 2023) finds grokking in a modular arithmetic task with curves that look very much like (5b) as well.
>
> In the new Appendix Section 17, we reproduce grokking on a modular arithmetic task (the task in Power et al, 2022) and show the striking accuracy curves we associate with grokking in Figure 20 and note the following facts
> * The test loss is decreasing before train loss is zero
> * The test loss saturates to a nonzero value at late time while the test accuracy reaches $100\%$.
>
> This shows that the types of loss curves we document in our experiments are actually *the same kind of loss curves* as those in original grokking experiments.
>
> We thus disagree with the claim that the original accuracy-based result from Power et al. is stronger (more surprising) than the corresponding loss curves that we treat here, and in fact they are equivalent. As we expect, our conclusions about grokking in loss naturally carry over to grokking in accuracy (but not vice versa). This is why when we intervene on classification-first tasks like modular arithmetic (Fig 2, Fig 6b) and MNIST image classification (Fig 6a), we see our interventions control grokking *exactly as we predicted*. If our definition (reasoning about loss) was a weaker one, we would not see this in classification tasks, where accuracy is plotted.
>
> We end by noting that we are not the first to make this observation. Indeed, the fact that loss, not accuracy, is the interesting thing to study was quickly realized after the initial paper discovering grokking was published! It is noted when (Davies et al, 2023) study loss in their Figure 5b, when (Liu et al, 2022b) characterize the *loss* landscapes of various tasks that exhibit grokking, when (Nanda et al, 2023) plots excluded and restricted loss as progress measures in and remarks that accuracy can be gamed, and strikingly in the recent (Schaeffer et al, 2023) that shows how accuracy and other metrics that are not optimized by the network can be very misleading to plot and easily gamed. Thus our choice to do so in this paper is consistent with past literature on grokking. We refer you to Appendix Section 17 created specifically to clarify this perceptive question you raise.
>
>
> **End-to-End Theory of Grokking**
>
> While the original paper did not attempt to provide an end-to-end theoretical description of the grokking dynamics, we now include a dynamical mean field theory (see the new Section 5.3 and Appendix 13) for networks trained on high dimensional data. In particular, we can write down simple expressions for the gap between test and train losses as functions of time (see equation 10) which states the test train loss gap at large $\gamma$
>
> $$\mathcal L_{test} - \mathcal L_{train} \sim \frac{2\alpha}{N \gamma} \sum_{ij=1}^N \int_0^t ds R^w_{ij}(t,s) C_{ij}^g(t,s) + \mathcal{O}(\gamma^{-2})$$
> where $R^w$ and $C^g$ are certain functions which can be computed from our theory. The test and train loss dynamics at large width in the lazy regime should emerge as a special case of this more general theory.
>
> **First Paragraph of Introduction Sounds Slightly Wrong**
>
> You make good points. We have clarified what we meant and rephrased the introductory paragraph. We now elaborate on the points we makes to set up the background motivation for the paper, notably with gentler language than before.

---

> > ### Author Response · Authors · 2023-11-16
> > **Response Part 3**
> >
> > **Comparison to Li et al**
> >
> > While we appreciate the contributions of Li et al on the implicit bias of SGD dynamics with label noise, and on lower bounds for kernel generalization error, we point out the following novel contributions in our work. First, our results and experiments do not rely on either SGD or on label noise, so explanations which rely on these phenomena are insufficient to explain our setting.  We also believe our conceptual contributions go beyond that analysis in this new setting studying grokking
> > 1. Studying how rapid feature learning can reduce the gap in time between train loss and test loss decreases
> > 2. Showing how harder tasks for the initial kernel lead to more extreme grokking.
> > 3. Providing an asymptotic description of networks (their weights, activations, train and test loss) trained on random data *far from the kernel regime* using DMFT techniques. We have provide an approximate equation for the test and train loss gap valid for large but finite $\gamma = P/D$ (see Equation 7).
> >
> > We hope in light of these improvements and contributions, the reviewer will consider increasing their score. We thank you for your detailed comments.

---

> > > ### Author Response · Authors · 2023-11-21
> > >
> > > We are writing again to ask if the reviewer thought our rebuttal addressed their questions and concerns or if there is anything else left for us to clarify before the response period ends. Also, we were wondering the new theoretical contributions, clarification of why loss is a more informative metric than hard accuracy, and the discussion of the work of Li et al the reviewer would consider increasing their score. Thank you for your time.

---

> > > > ### Comment · Reviewer_y5Qp · 2023-11-22
> > > >
> > > > Thank you for your response! I appreciate the comparison to Li et al. Since the authors have revised the paper substantially, it would be helpful if the authors could mark the major edits in a different color.
> > > >
> > > > I am not very familiar with DMFT and haven't had time to read the added theoretical analysis carefully. I will need to discuss this with other reviewers to better assess the contribution of the new theoretical results.  Here is a quick question I have on the new theoretical result, to me any theory showing grokking needs to prove there exists two timesteps $t_1< t_2$, where
> > > >
> > > > 1. the training loss/error is almost 0 at time $t_1$ while the test loss/error is high (larger than some constant) at time $t_1$;
> > > > 2. both train and test loss/error is almost 0 at time $t_2$.
> > > >
> > > > However, I cannot find such statements in the DMFT section. Can the authors elaborate on this?

---

> > > > > ### Author Response · Authors · 2023-11-22
> > > > >
> > > > > We thank the reviewer for the question.
> > > > >
> > > > > **Revised and new sections now in blue**
> > > > >
> > > > > We have changed the color of the modified sections (Section 5.3, Appendix Section 13, Appendix Sections 14-18) to blue to make it clearer what the new additions are.
> > > > >
> > > > > **What is DMFT?**
> > > > >
> > > > > Dynamical mean field theory is a method for obtaining the train and test loss dynamics for high dimensional data and large datasets. The theory operates in the limit of $P,D \to \infty$ with $P = \gamma D$. The summary statistics in equation 4 concentrate in this limit and characterize the train and test loss dynamics, from which we can predict grokking behavior.
> > > > >
> > > > > **The need for dynamics: motivating DMFT**
> > > > >
> > > > > Thank you for your theoretical question about timesteps, it cuts to the heart of the issue. The two conditions you ask from a theory of grokking can be restated as saying a theory of grokking must demonstrate there exist $t_1<t_2$ so that:
> > > > > 1. $L_{test}(t_1)-L_{train}(t_1) \gg 0$ is large with $L_{\text{train}}(t_1) \approx 0$ small (a time at which we memorize but do not generalize).
> > > > > 2. $L_{\text{test}}(t_2) \approx L_{\text{train}}(t_2) \approx 0$ is small (end-time generalization). Notice to show this it suffices to show $L_{\text{test}}(t_2)-L_{\text{train}}(t_2) \approx 0$ and $L_{\text{test}}(t_2) \approx 0$.
> > > > >
> > > > > The fact that we want to reason about this test-train gap *at different points in time* means that it does not suffice to prove facts about *static* quantities like lower bounds on end-time test loss, etc, and that we must reason about various quantities (weights, activations, this test-train gap) *dynamically*. DMFT is just a common technique [(Mignacco et al., 2020; Celentano et al., 2021)] to do exactly this. Note we can only use this mathematically involved technique to reason this way because our setting is as simple as it is: doing theory on dynamics for more complicated models like one-layer Transformers (Nanda et al, 2023) or MLPs with AdamW (Liu et al, 2022b) is untenable, for instance.
> > > > >
> > > > > **Figure 4 exactly demonstrates the existence of $t_1, t_2$ is predicted by our theory**
> > > > >
> > > > > Consider Equation (7):
> > > > >
> > > > > $$L_{\text {test }}(t)-L_{\text {train }}(t) \sim \frac{2 \alpha}{\gamma N} \sum_{i j=1}^N \int_0^t d s R_{i j}^w(t, s) C_{i j}^g(t, s)+\mathcal{O}\left(\gamma^{-2}\right)$$
> > > > >
> > > > > From this, we can make observations about $t_1, t_2$ at two levels of abstraction
> > > > >
> > > > > - See how the test-train gap dynamics scale qualitatively with $\gamma, \alpha, \epsilon$, to predict what tuning of these parameters gives grokking. On page 8, the "Overfitting effects at finite $\gamma$" and "Effect of $\epsilon$ and $\alpha$" do exactly this, and experiments (eg. Figure 3) to verify these insights.
> > > > > - **DMFT outputs implicit integral equations like (7) that can be solved numerically. We want to emphasize the DMFT plots in Figure 4 are the predictions of our theory, so the existence of a test-train gap in those curves obeying conditions (1, 2) from earlier means the required $t_1, t_2$ exist and can be read off the curves. For instance, Figure 4(a)(right) shows how $\gamma = 15$ has $t_1 = 20, t_2 = 50$ where initially train loss halves before test loss starts moving at all. This would become even more emphatic for smaller $\gamma$ (ie. train loss would go near zero before test loss starts moving -- exactly what is demanded in conditions (1,2)), but solving the DMFT integral equations is numerically challenging there. Similarly, in Figure 4(b)(right) for $\alpha = 10^{-1}$, we see $t_1 = 18, t_2 = 60$ satisfies the two conditions also.**
> > > > >
> > > > > The reviewer might be wondering about simple, closed-forms for $t_1, t_2$. We address this next.
> > > > >
> > > > > **Cannot get $t_1, t_2$ in simple closed form because train and test loss dynamics are computationally hard**
> > > > >
> > > > > We cannot get $t_1, t_2$ in simple closed-form, *because knowing test and train loss at every $t$ is, in some sense, at least as computationally hard as running gradient descent.* The key is that DMFT outputs *implicit* equations characterizing loss over time, hence the need to solve them numerically. More specifically, it is hard to calculate simple closed-forms for the test-train gap because because the final answer depends on solutions to self-consistent equations for the summary statistics defined in (4), which are computed as averages over non-Gaussian stochastic processes (5-6). Thus it is challenging to "read-off" timescale gaps simply by inspecting the integral equations alone (without solving them numerically).
> > > > >
> > > > > We thank the reviewer for their question and hope this illustrates how our theory of grokking does indeed predict such a $t_1 < t_2$ when its final equations are (numerically) solved out, as desired. The key Figures illustrating this are Figure (4) and Figure (13).

---

### Official Review · Reviewer_Hhk6 · 2023-11-07

**Soundness:** 2 fair
**Presentation:** 2 fair
**Contribution:** 2 fair
**Rating:** 5
**Confidence:** 3

**Summary:**

In this submission, the authors proposed a simple regression model that exhibits the grokking phenomenon, that is, good generalization is achieved long after the decay of training error. The studied model is a polynomial regression problem, where the teacher model is a quadratic single-index model, and the student model is a neural network with degree-2 activation function. It is predicted that under suitable training set size and initialization scale, grokking occurs in the intermediate phase between the NTK regime and the feature learning regime.

**Strengths:**

The proposed toy model is interesting and relevant to the ICLR community. Most existing works on grokking considered modular arithmetic tasks, or measured the classification error rather than the surrogate loss used in training. In contrast, the authors studied the regression setting where grokking is manifested in the $L_2$ error, and the training procedure does not involve $\ell_2$ regularization. Also, the connection between grokking and the transition from lazy to rich regime is to my knowledge novel, and it opens up new directions for theory research to rigorously characterize such phenomenon.

**Weaknesses:**

I have the following concerns.

1. Given the idealized setting (Gaussian data with identity covariance, single-index target), it is rather underwhelming that the authors did not provide any quantitative characterization of the training dynamics to prove the existence of grokking. Instead, the proposed explanation is only verified empirically, which limits the contribution. Can the authors comment on the technical challenges in analyzing the gradient flow trajectory for this quadratic model?

2. Connection to prior results needs to be further elaborated.

* (Nichani et al. 2022) also studied the transition from the NTK to the feature learning regime, and established improved sample complexity for learning low-degree + sparse high-degree polynomials. For the current model, does the sparsity of the higher-order term (that is, whether the quadratic component of the target function span more than one direction) play a role in the presence of grokking?
(Nichani et al. 2022) *Identifying good directions to escape the NTK regime and efficiently learn low-degree plus sparse polynomials.*

* The target function is a single-index model with information exponent 2 (Ben Arous et al. 2021). It is known that for such target function, gradient descent on the population loss will start from an approximate saddle due to the random initialization. Is this observation related to the test loss plateau in grokking? And do we expect similar findings when the information exponent is larger than 2?
(Ben Arous et al. 2021) *Online stochastic gradient descent on non-convex losses from high-dimensional inference.*

* The quantity describing the difficulty of target function $y^\top K^{-1} y$ has been extensively studied in the kernel literature. See (Arora et al. 2019) for the analysis of NTK.
(Arora et al. 2019) *Fine-grained analysis of optimization and generalization for overparameterized two-Layer neural networks.*

* (minor) The cited references on the quadratic sample complexity lower bound for kernel estimators such as (Ghorbani et al. 2019) do not rigorously cover the case of Gaussian data.

**Questions:**

See Weaknesses above.

---

> ### Author Response · Authors · 2023-11-16
> **Response Part 1**
>
> We thank the reviewer for their indepth reading and their useful suggestions. We took their suggestion to provide a more predictive theoretical description seriously and added a whole section dedicated to this. We also investigated multi-index and higher degree hermite polynomial target funtions.
>
> Below we respond to each of the weaknesses.
>
> Weaknesses:
>
> 1. *Given the idealized setting (Gaussian data with identity covariance, single-index target), it is rather underwhelming that the authors did not provide any quantitative characterization of the training dynamics to prove the existence of grokking. Instead, the proposed explanation is only verified empirically, which limits the contribution. Can the authors comment on the technical challenges in analyzing the gradient flow trajectory for this quadratic model?*
>
> We had begun deriving the dynamics of the test risk such a model, but did not include it in the initial submission because solving the resulting equations numerically was challenging. However, the reviewer's questions about theory have prompted us to complete and include this theoretical work that can potentially shed light on where grokking comes from in terms of data and richness. We believe this substantially strengthens the submission.
>
> In particular, we now derive a dynamical mean field theory (DMFT) description of the dynamics in the proportional regime where $P = \gamma D$ with $P ,D \to \infty$ (with some subtleties about initial conditions for the alignment of weights to target direction). See Section 5.3, Appendix 13 and Figure 4.
>
> This results in a version of the classic Saad-Solla [1] equations  for two-layer network dynamics, but with data repetition which induces correlations in weight updates and preactivations across time (allowing for the possibility of overfitting/grokking).
>
>
> The key mathematical challenges that arise from having a separate train/test distribution when fitting a nonlinear function in this high dimensional limit are
> 1. Non-markovian dynamics for the weights and preactivations (see equations 7-8), which give compouned errors between the online ($\gamma = \infty$ dynamics) and offline ($\gamma$ finite) solutions. (For example, see the matrix $R^h$ in Figure 13d which is no longer diagonal!)
> 2. Non-Gaussian preactivation statistics (see equation 6).
>
> However, we can still recover interesting dependences on $\alpha$, $\gamma$, and $\epsilon$ (see Figure 4). For example, reducing $\alpha$ and increasing $\gamma$ reduces the gap between train and test loss dynamics.
>
> \begin{equation}
> \mathcal L_{test}(t) - \mathcal L_{train}(t) \sim \frac{2 \alpha}{\gamma N} \sum_{ij=1}^N \int_0^t ds R^w_{ij}(t,s) C^g_{ij}(t,s) + \mathcal{O}(\gamma^{-2})
> \end{equation}
>
>
> where $C^g$ is the gradient correlation function and $R^w$ is the response function for the weight dynamics. All of these functions implicitly depend on $\gamma, \epsilon,\alpha$. The DMFT allows us to recover from a theoretical framework, the phenomena we observed empirically in our polynomial experiments.
>
> [1] https://journals.aps.org/pre/abstract/10.1103/PhysRevE.52.4225
>
> 2. *(Nichani et al. 2022) also studied the transition from the NTK to the feature learning regime, and established improved sample complexity for learning low-degree + sparse high-degree polynomials. For the current model, does the sparsity of the higher-order term (that is, whether the quadratic component of the target function span more than one direction) play a role in the presence of grokking?*
>
> There does not seem to be a major change in the grokking dynamics when we fit multi-index functions compared to single-index functions. We've added a new set of experiments in Section 15.1 of the Appendix verifing that the grokking we present holds out-of-the-box on multi-index models where the quadratic component spans multiple directions, illustrating that sparsity is not necessary for our results to hold.
>
> We thank you for bringing (Nichani et al, 2022) to our attention. It is a very pertinent example of a transition from lazy to rich dynamics and what kinds of feature learning corrections improve generalization when transitioning away from the NTK. We added a citation to this work.

---

> ### Author Response · Authors · 2023-11-16
> **Response Part 2**
>
> 3. *The target function is a single-index model with information exponent 2 (Ben Arous et al. 2021). It is known that for such target function, gradient descent on the population loss will start from an approximate saddle due to the random initialization. Is this observation related to the test loss plateau in grokking? And do we expect similar findings when the information exponent is larger than 2?*
>
> Indeed, we believe these phenomena are related. In Appendix Figure 10, we exactly tested the effect of "warm start" initial conditions where the weights are partially aligned to the target direction. All things equal, the models with poor alignment (close to saddle point of dynamics) exhibit more dramatic grokking behavior. Conversely, weight initializations with some component of the solution weights to begin with (far from saddle point) have train and test curves that track each other more strongly over the course of training, not exhibiting grokking to the same degree.
>
> You also make a good point about the fact that the setting with information exponent $k=2$ is perhaps an easier task (in the sense of escaping the saddle point at initialization) than the case where $k \ge 3$, as (Ben Arous et al, 2021) outline in Theorem 1.3 and 1.4. Thus, it is natural to ask whether the grokking that arises does so because of any properties related to the quadratic target. We can test this by seeing if grokking arises for $k \ge 3$ higher degree Hermite polynomials. We find that indeed it does, out-of-the-box, so we can be confident that the findings do not rely on special properties of the quadratic target (information exponent $k=2$). We have new experiments illustrating this in Section 15.2 of the Appendix.
>
> 4. *The quantity describing the difficulty of target function $y^T K^{-1} y$ has been extensively studied in the kernel literature. See (Arora et al. 2019) for the analysis of NTK. (Arora et al)*
>
> We have updated the paper to emphasize that this is not a novel metric we are proposing, but rather is a natural, well-studied, object which reflects the RKHS norm of the targets on the train distribution, and turns out to be useful in the examination of grokking. We now cite Arora et al and other appropriate papers when introducing this difficulty metric.
>
> 5. *(minor) The cited references on the quadratic sample complexity lower bound for kernel estimators such as (Ghorbani et al. 2019) do not rigorously cover the case of Gaussian data.*
>
> This is a good point, we have updated the paper to qualify these references when we point to them.
>
> Thank you for the detailed comments! We believe the additional theory (DMFT for dynamics) and experiments (new Appendix Sections 15-18) substantially improved the paper as now the emergence of a test-train gap is theoretically justified in addition to empirically controlled by the parameters we introduce. We hope that the reviewer will consider raising their score based on the new theoretical contributions, and additional Appendix sections.

---

> > ### Author Response · Authors · 2023-11-21
> >
> > We are writing again to ask if the reviewer thought our rebuttal addressed their questions and concerns or if there is anything else left for us to clarify before the response period ends. Also, we were wondering if in light of the new results, empirical tests and reorganization of the theory section of the paper, the reviewer would consider increasing their score. Thank you for your time.

---

### Author Response · Authors · 2023-11-16
**Global Response**

We thank all of the reviewers for their in-depth reading and useful comments. As a direct result of responding to reviewer questions, we have added over 10 pages to the manuscript, mostly in the Appendix but also modified the main text. This now includes
- Theory for dynamics of test risk (in the form of Dynamical Mean Field Theory equations) that takes advantage of the unusually simple setting in which we observe grokking, giving theoretical insight into the data regime in which grokking occurs as well as new experiments to test this theory. We find very close agreement, which allows us to then inspect the derived equations governing the dynamics of weights and activations, finally resulting in a clean interpretation for how the parameters of interest ($\gamma, \alpha, \epsilon$, see below) control the gap between train and test loss, and thereby the mechanism by which they induce (and can eliminate) grokking. In particular, Equation (7) resulting from this DMFT gives a quantitative, theoretical characterization of the test-train gap and its dynamics. This new theory can be found in the new Section 5.3 of the main text, and Appendix Section 13. To make space for this new theory in the main text, some things like the empirical loss decomposition curves and use of small eigenvectors to induce grokking on Gaussian data, have been moved to the Appendix.
- New experiments on variants of our polynomial regression tasks, including multi-index models and higher degree Hermite polynomials, verifying they show grokking consistent with our hypothesis and illustrating that our thesis does not rely on any special properties of quadratic single-index models. This comprises the new Section 15 of the Appendix.
- A detailed discussion of grokking in loss vs grokking in accuracy in Appendix Section 16, and new experiments illustrating why loss, not accuracy, is a more informative metric to study in context of grokking. We believe these additions to the appendix both generalize and clarify our results, so we thank the reviewers for prompting us to include them as response to the reviewer comments and questions. We also added Appendix Section 17 to show how hard tasks (such as learning small eigenvectors of a data matrix) can be learned with enough data in the rich regime.

Some things we wanted to emphasize were, first, that the interesting part of grokking is that it is a *dynamical* phenomenon -- that is to say, the gap between train and test loss is one that is only present in the middle of training; a network that groks eventually generalizes so that train and test loss are both very low at end-time. This is why a theory of dynamics which allows for repeated use of training data points (DMFT) is necessary, and reasoning about end-time loss or test risk is not enough to understand grokking. Second, sometimes the use of toy models is a limitation of a work, but here, the fact that we can get grokking in a toy model is an important advantage, because it refutes prior theories of grokking [(Liu et al, 2022b), (Varma et al, 2023), (Thilak et al, 2022)] that posit more complicated mechanisms behind this empirical phenomenon.

**Can we develop more predictive theory?**

We have added a Dynamical Mean Field Theory (DMFT) asymptotic description for network dynamics when the hidden unit size $N = \mathcal{O}(1)$ and $P = \gamma D$ with $P, D \to \infty$ (see section 5.3 and Appendix 13). This theory gives the typical train + test losses when training on random data in high dimension. Of particular interest is an equation for the *gap* between test and training loss as a function of the problem parameters.

The general result is a set of coupled integro-differential equations which describe the distribution of preactivations $h=w \cdot x$ and weight dynamics. Though solving these equations is challeging, we show that this can capture the dynamics of our polynomial task (see Figure 4, 13). We show also that in the limit of $\gamma \to \infty$ our equations simplify dramatically and we recover the classic Saad-Solla [1] equations for online learning. Solving the DMFT equations in settings where grokking is most dramatic is still challenging, especially as finite size effects appear larger here (Figure 4c), but we hope this opens an avenue for future works to analyze grokking theoretically.

[1] Saad, Solla. Online Learning in Soft Committee Machines (1995).

*In response to reviewer y5Qp, we have changed the color of new/updated sections in the manuscript to blue. We also point any reviewers curious about DMFT, its motivation, role and consequences for grokking, to our most recent response to reviewer y5Qp, who asks about DMFT and the new theoretical results we have added to our paper in response to the reviews.*

---

### Meta-Review · Area_Chair_Spnx · 2023-12-10

**Metareview:**

In this paper the authors suggest that the grokking phenomenon, characterized by a neural network's training loss decreasing significantly earlier than its test loss, may occur when the network shifts from a phase of lazy training dynamics to a richer regime of feature learning. To demonstrate this concept, they examine they focus on the scenario of applying vanilla gradient descent to a polynomial regression problem using a two-layer neural network. This network displays grokking without the need for regularization, a behavior they claim is not accounted for by existing theories. In their study, they pinpoint specific statistics that are key to predicting the test loss of such networks. Monitoring these statistics throughout the training process, they observed that grokking in this context happens when the network initially attempts to fit a kernel regression solution using its initial features. This is followed by a later phase of feature learning, during which a solution that generalizes is found, even though the training loss is already low.

The reviewers liked the hypothesis that grokking happens as a transition from kernel regime to feature learning. However they did also raise major concerns about the strength of the empirical and theoretical evidence provided. During the response period the authors added a DMFT section to alleviate some of the concerns. One reviewer in particular raised important concerns however as they put it:

"Mathematically, the original grokking definition at least (let's ignore 'long after' and only require 'after' for now) requires there exists two time-steps t_1<t_2, where the training loss/error is almost 0 at time t_1 while the test loss/error is high (larger than some constant) at time t_1, both train and test loss/error is almost 0 at time t_2. However, in this manuscript, especially in the recently added DMFT section, the authors focus on whether there is some time-step t, where test loss - train loss is large at t. While I agree this paper has interesting observations and possibly interesting technical results in the DMFT section, I find the presentation of this paper very misleading. The authors should be frank with this different and relaxed definition of grokking, which is probably one of the main difficulties in coming up with a rigorous theory even on simple settings to show grokking as a transition from kernel to rich regime for l2 regression."

The more positive reviewers agreed for the most part that this is an important shortcoming and a relaxed definition of grokking but asserted that perhaps this more relaxed definition is more widely observable. All reviewers also agreed that the new section DMFT dynamics "introduce a lot of complexity to the paper without a huge payoff, as the discussion and interpretation of the resulting dynamics is quite limited/not that enlightening." The positive reviewers decided to keep their scores based on the empirical evaluations.

It is clear that this paper has some interesting ideas and so keeping it in the program I think will benefit the community. That said there are clearly misleading and not substantiated claims in the paper. Therefore I recommend that the authors thoroughly revise the paper befor publishing and in particular I would strongly recommend removing the DMFT part.

**Justification For Why Not Higher Score:**

There are clear concerns about this paper as I have outlined in my meta review. While I think the benfits outweighs the flaws clearly not spotlight or oral worthy IMO

**Justification For Why Not Lower Score:**

This paper is borderline but the experimental observation and the view of groking as transition from lazy to rich is interesting and the community will benefit from further discussion around this paper.

---

### Decision · Program_Chairs · 2024-01-16

Accept (poster)